# Wood Vinegar Promotes Soil Health and the Productivity of Cowpea

Edwin K. Akley [1,*], Peter A. Y. Ampim [2], Eric Obeng [2], Sophia Sanyare [1], Mawuli Yevu [1], Eric Owusu Danquah [3], Ophelia Asirifi Amoako [1], Theophilus K. Tengey [1], Justice K. Avedzi [4], Vincent K. Avornyo [4], Abdul Fatawu Neindow [1] and Abdul Fatawu Seidu [5]

[1] Council for Scientific and Industrial Research (CSIR)-Savanna Agricultural Research Institute (SARI), Tamale P.O. Box TL 52, Ghana; sanyaresophia29@gmail.com (S.S.); mawuliyevu@yahoo.com (M.Y.); opheliaasirifiamoako@gmail.com (O.A.A.); tktengey@gmail.com (T.K.T.); neindowabdulfatawu@gmail.com (A.F.N.)

[2] Department of Agriculture, Nutrition and Human Ecology and Cooperative Agricultural Research Center, College of Agriculture, Food and Natural Resources, Prairie View A&M University, P.O. Box 519, MS 2008, Prairie View, TX 77446, USA; paampim@pvamu.edu (P.A.Y.A.); erobeng@pvamu.edu (E.O.)

[3] Council for Scientific and Industrial Research (CSIR)-Crop Research Institute, Kumasi P.O. Box 3785, Ghana; e.owusudanquah@cropsresearch.org

[4] Department of Soil Science, University for Development Studies, Tamale P.O. Box TL 1882, Ghana; javedzi@uds.edu.gh (J.K.A.); vavornyo@uds.edu.gh (V.K.A.)

[5] Department of Agriculture, Tolon District Assembly, Tolon P.O. Box TN 4, Ghana; saeedafatahi8@gmail.com

[*] Correspondence: akleykorbla@gmail.com; Tel.: +233-24-2117607

**Abstract:** Wood vinegar (WV) is a biostimulant and a biopesticide that contains pyroligneous acid and is used as a crop growth enhancer and biocontrol agent, but insufficient information exists on WV's effects on soil quality and cowpea production in Ghana. A field study (2 years) was conducted to determine the appropriate method of applying WV for soil health and cowpea production, and to determine the economic benefits of WV. Assessments were on nodulation, shoot biomass, yield, value–cost ratio, soil enzymes, soil respiration, microbial biomass nitrogen (MBN), permanganate-oxidizable carbon (POXC), mineralizable C, soil pH, available nitrogen and phosphorus. Results revealed soil drenching and foliar application are efficient methods of applying WV to achieve greater shoot dry matter, nodulation and grain yield of cowpea. Economically, soil drenching, followed by foliar application, generated better economic returns than the control. Adding WV to cowpea using soil drenching and foliar application improved soil health indicators. Soil enzymes and MBN were enhanced by WV applied foliarly and through soil drenching. Soil drenching with WV produced greater POXC and mineralizable C compared to the other treatments. Conclusively, WV applied foliarly and through soil drenching improved soil health, nodulation and yields of cowpea, and enhanced profitability.

**Keywords:** pyroligneous acid; soil quality; soil enzymes; cowpea grain yield; soil drenching; foliar application; value: cost analysis



## 1. Introduction

The sustainable intensification of agricultural practices that focus on promoting innovations that enhance crop productivity without altering the ecosystem services of the soil and the environment are crucial in the face of climate change. Wood vinegar, a product of pyroligneous acid, is receiving a significant attention in sustainable agricultural systems because of its multi-purpose uses [1]. Wood vinegar is a byproduct obtained from the pyrolysis of plant biomass to produce biochar [1]. The thermal carbonization of the plant biomass ends in biochar, condensable products (tar and pyroligneous acid) and other non-condensable gaseous products ($H_2O$, $CH_4$, $H_2$, $CO_2$, CO) [2]. The quality of pyroligneous acid (PA) depends on the quality of wood species used for pyrolysis [3,4]. Pyroligneous acid (PA) is made up of over 200 water-soluble chemical compounds (80–90% with water)

and organic chemical compounds (10–20%) [3]. The composition of the organic compounds includes organic acids, alkanes, phenolic compounds, alcohol and esters, of which acetic acid accounts for about 50% in solution [3].

The benefits of PA have been extensively documented by several authors. Pyroligneous acid has been used as a foliar fertilizer, a plant growth enhancer, an additive to compost, a repellent for pests and disease, an antimicrobial and an antioxidant in conventional agricultural systems [5,6]. Lately, PA is attracting significant interest in organic crop production systems because of its use as a natural biofertilizer and biocontrol agent for pests and diseases [7]. As a plant growth enhancer, PA increases crop resistance to stress, strengthens photosynthesis in crops to promote crop growth and yield, and enhances nutrient absorption through the roots [8]. Thus, it can be a safer alternative to chemical fertilizers which pose threats to the environment. Lashari et al. [8] observed that a combined application of organic fertilizer (biochar–manure compost) and PA improved the growth and yield of maize as well as reduced salt stress to plant in saline soils.

Regarding soil quality improvement, pyroligneous acid can positively affect soil quality [6,9] and, when correctly applied, can produce substantial economic and environmental benefits [1,10]. Thus, it can be an effective soil improver [10]. The addition of PA has been found to improve the availability of N, P, and K in the soil [11] and positively alter soil pH [10]. Likewise, Seo et al. [12] observed that combined herbicide–pyroligneous acid mixtures improved soil acidity, organic matter, available P, and electrical conductivity (EC) and subsequently contributed to higher residual nutrients in the soil. The application of PA at an optimal rate has been found to help mitigate nitrous oxide ($N_2O$) and methane ($CH_4$) emissions in N-fertilized rice paddy soil [11] and also inhibited ammonium loss [3]. A significant improvement was observed in soil salinity, soil pH, and bulk density in plots treated with combined organic fertilizer and PA compared with those under conventional practices [8]. Additionally, Cardelli et al. [5] reported that the application of a low concentration of PA (only up to 1% dilution dose) to the soil had a positive influence on the soil microbial community. Hence, PA can potentially improve soil biological fertility. Nonetheless, Cardelli et al. [5] recommended that further studies are needed to investigate the influence of PA on soil and crop production. In addition, Grewal et al. [3] reported that there is inadequate scientific research on PA's effects on soil health.

In Ghana's farming systems, using WV or PA to enhance crop productivity and soil health is a new technology. Wood vinegar technology is being recommended as a biofertilizer for stimulating crop growth and development and as a biocontrol agent for the control of pests and diseases. However, no attempt has been made to investigate the efficacy of WV application methods and rates on crop production, limiting the development of WV technology in enhancing crop productivity and restoring soil health. Using WV as a biofertilizer for crops can alter soil biological and biochemical properties because such properties are associated with microbial activities (nutrient cycling) and are sensitive to short-term changes in management. Thus, manipulating soil quality through the correct use of WV for crop production can be a sustainable agricultural practice for enhancing food security and environmental quality in smallholder farming systems in Ghana.

Cowpea [*Vigna unguiculata* (L.) Walp.] is an important legume grown primarily by smallholder farmers because its grain and biomass are highly valued for food and forage in Ghana. Cowpea is regarded as "the poor man's meat" because of its high nutritional food value for humans and animals. It improves soil quality through fixing nitrogen symbiotically with soil rhizobia, and up to 200 kg N ha$^{-1}$ has been reported as $N_2$ fixed by cowpea [13]. Hence, it forms an important component of smallholder farming systems in the northern part of Ghana. Although cowpea is becoming an important economic (cash) and social crop in Ghana, its yields are low (<1 t ha$^{-1}$) on smallholder farms due to poor soil fertility, the low adoption of best management practices, and the high cost of quality agro-inputs [14].

The adoption of WV or PA technology in low-external inputs cowpea smallholder farming systems, thus, can be a potential sustainable intensification practice to reverse

declining soil quality and to improve cowpea productivity and the livelihood of smallholder farmers in the northern part of Ghana. Previous studies have proven that WV or PA application enhanced the growth and yields of legumes in regions of Asia [15–17]. However, in Ghana, and by extension many sub-Saharan West African countries (SSWA), the effect of WV or PA on cowpea production and soil health have not been investigated well. Meanwhile, the current demands for cowpea in local and international markets have created opportunities to improve the productivity of existing climate-smart improved cultivars through leveraging emerging cost-effective technologies such as the use of WV.

The present study, therefore, hypothesized that adopting the correct timing/frequency and method of applying WV will improve soil quality, resulting in increased cowpea plant vigor and grain yield. Thus, the overarching objective of this paper is to determine the effect of the correct application (method) of WV on selected soil health indicators. The specific objectives were to determine the effect of different methods of WV application on (i) soil biological and chemical properties, (ii) the nodulation and yield of cowpea and (iii) the value–cost ratio resulting from the use of WV.

## 2. Materials and Methods

### 2.1. Study Site, Cropping History, Experimental Design and Treatments

The study was conducted at the Council for Scientific and Industrial Research (CSIR)—Savanna Agricultural Research Institute (SARI) field located in Nyankpala (N 9°23′36.6768″, W 1°0′16.90012″ in 2021 and N 9°23′36.558″, W 1°0′4.332″ in 2022) in the Northern Region of Ghana during the 2021 and 2022 cropping seasons. The main vegetation found in the study area is grassland (typical savanna) with a few clusters of drought-resistant trees such as African baobabs (*Adansonia digitata* L.), acacias [*Acacia nilotica* (L.) Willd. ex Delile], African locust bean (*Parkia biglobosa* Jacq.) and shea nut tree (*Vitellaria paradoxa* C.F. Gaertn.) [18]. The area has a unimodal rainfall pattern with an annual mean ranging between 750 and 1050 mm. Rainfall distribution persists for 5–6 months annually, with peak rainfall occurring in July to September, whereas temperatures vary between 14 °C at night and 40 °C during the day. The mean relative humidity (RH%) from May to December varies between 60% and 80% [18].

Information on the weather at the experimental site has been provided in supplementary Table S1. The data on daily temperature and rainfall distribution as well as humidity were sourced from the Agro-meteorological Section of the CSIR-SARI. Briefly, in 2021, the peak mean rainfall was experienced in April, and rainfall distribution lasted for 9 months, whereas in 2022, the peak mean rainfall was observed in August, and rainfall distribution lasted for 8 months. In general, rainfall was higher in 2021 than in 2022 (Table S1). Thus, rainfall data suggest that 2022 was a drier year relative to 2021. Similarly, the mean monthly temperature revealed that 2021 was a bit warmer compared with 2022 (Table S1). From January–April, mean temperatures were 31 and 30 °C for 2021 and 2022, respectively, indicating warmer temperature for 2021. This period (January–April) coincided with the "harmattan season" (period characterized by drought spell). Likewise, between August and October, when cowpea production is mostly carried out, mean temperatures for 2021 were still higher compared to 2022 (Table S1). On the other hand, mean relative humidity data indicated that 2022 had higher mean relative humidity (RH%) relative to 2021 (Table S1).

The field was disc ploughed before baseline soil samples ($n = 10$) were collected from 0–15 cm depths, air-dried and passed through a 2 mm sieve to remove debris before baseline nutrient characterization was performed using the procedure of Akley et al. [19] and Ulzen et al. [20]. Baseline nutrient characterization is provided in Table 1.

**Table 1.** Baseline soil nutrient characterization for 2021 and 2022 cropping seasons.

| Parameters (0–15 cm) | 2021 | 2022 |
| --- | --- | --- |
| | Value | Value |
| Soil Types (USDA Classification) | Typic Plinthustalf | Typic Plinthustalf |
| SOC (Walkley–Black procedure), g kg$^{-1}$ | 3.19 | 4.80 |
| Total N (Kjeldahl procedure), g kg$^{-1}$ | 0.37 | 0.44 |
| Available P (Bray-1), mg kg$^{-1}$ | 2.78 | 0.96 |
| Available N (NH$_4^+$-N + NO$_3^-$-N), mg kg$^{-1}$ | 0.21 | 0.13 |
| Soil pH (soil: water; 1:5) | 5.7 | 6.00 |

The 2021 experimental site had a cropping sequence of soybean–maize–fallow in 2018, 2019 and 2020, respectively, whereas the 2022 experimental field had a cropping management history of soybean–fallow–maize in 2019, 2020 and 2021, respectively.

The experiment was set up as a randomized complete block design (RCBD) with four replications per treatment. The treatment consisted of three methods of WV application, namely, a control (con), soil drenching (SD) and foliar application (FA). Each experimental plot measured 3.6 m × 3 m. The experimental plots were manually ridged with hoes at 60 cm apart, and each plot had a total of six ridges.

A commercial wood vinegar branded as an organic farming aid (OFA) was used for the experiment. OFA wood vinegar is produced and marketed in Ghana by HJA Africa, located in the premises of CSIR-Industrial Research Institute in Accra, Ghana. OFA wood vinegar contains 100% pyroligneous acid (PA), of which 2.3% is acetic acid, 0.52% is methanol, 0.5% is potassium, 0.001% is phosphorus and 0.01% is total N. The manufacturer recommends using the product in the open field via foliar application and drenching at a 0.2% (1:500) dilution on arable crops using tap water on a weekly or biweekly basis.

### 2.2. Planting, Agronomic Management and Sampling of Rhizospheric Soil

The cowpea seeds (*Vigna unguiculata* L. Walp. cultivar Wang-kae) used as the test crop were sourced from the cowpea improvement program section of the CSIR-Savanna Agricultural Research Institute (SARI) in Nyankpala, Ghana. Cultivar Wang-kae is a dual-purpose cowpea with maturity period of 70 days. It is resistant to Aphids (*Aphis craccivora*) and Striga (*Striga gesnerioides*). The cowpea was manually planted under rainfed condition at an inter-row (ridge) and intra-plant distance of 60 × 20 cm, respectively, on 16 August 2021 and on 10 August 2022. Three seeds were planted per hill and thinned to two stands per hill 10 days after planting (DAP).

At 25 days after planting (DAP) (i.e.,V4 stage), the WV treatments were applied using 45 mL WV dissolved in 22.5 L of tap water per treatment at an application rate of 300 mL ha$^{-1}$. The sprayer was calibrated prior to use for each WV application. For foliar application, the WV was sprayed directly on the leaves of the plants with the aid of a knapsack sprayer, while soil drenching was performed through applying the WV directly to the soil surface (near the rooting zone) using a knapsack sprayer. The control plots were treated with just water. After the first WV application, the first sampling (time zero sampling) was performed, which corresponds to the V4 stage. Subsequently, the WV was applied every seven days, while the sampling of plant biomass and rhizosphere soils was carried out every 14 days.

Rhizosphere soil samplings were performed at 25, 39, 53 and 67 DAP in 2021 and 25, 41, 53 and 67 DAP in 2022, which corresponds to the V4 (4-leaf), V8 (8-leaf), R2 (full flower) and R4 (full pod) stages, following the procedure of Hungria et al. [21]. Briefly, at each sampling stage, six (6) plants (outside the four central rows) were gently uprooted per plot with a spade and the soil around the roots was gently shaken off. Soil that adhered to the roots after the gentle shake-off was considered rhizosphere soil and was collected into a sampling bag and transported to the soil microbiology laboratory (Lab) on ice packs. At the Lab, soil samples that were collected from the rhizosphere during each sampling stage were

air-dried for 72 h, homogenized and sieved with a 2 mm mesh sieve to remove large plant materials and other debris before chemical and microbiological analyses were performed.

Weeds were controlled manually with hoes when necessary. Insects were controlled using K-Optimal (Lambda Cyhalothrine 15 g/L + Acetamipride 20 g/L EC) insecticide at 30 mL in 15 L of water and applied with a knapsack sprayer at an application rate of 300 mL ha$^{-1}$. The chemical control of insects began at the full flower to early pod stage (R5 stage) and was performed weekly for 3 consecutive times. Later, the spraying frequency increased to every three (3) days to prevent insect infestation in the pods. However, the insecticide application stopped at the full pod maturity stage. In all, six spraying regimens were applied.

### 2.3. Sampling and Data Collection

#### 2.3.1. Biomass Yield and Nodulation

Data on biomass (shoot and root dry matter) yield and nodulation (nodule mass) were collected from the six (6) plants sampled for rhizosphere soil per growth stage.

Briefly, at each rhizosphere soil sampling stage, the six (6) plants sampled per plot (outside the four central rows) were collected and put together, and then the roots with nodules were removed from the plants, leaving the shoots. Nodules that fell from the roots during the rhizosphere soil sampling were also collected. Later, the shoots and the roots with nodules were put into paper bags and transported to the Lab. In the Lab, the shoots and roots with nodules were washed under running tap water and air-dried for 30 min. The nodules were then removed from the roots before nodulation assessment was performed. Nodule mass was determined through drying the detached nodules in a forced-air oven (101-3AB Dry Oven Haunghua faithful Co. Ltd., Cangzhou, China) at 70 °C for 72 h and then weighed. Nodule mass was expressed on a per-plant basis in milligrams (mg plant$^{-1}$). Similarly, fresh biomass (shoots and roots) was dried in a forced-air oven at 70 °C for 72 h and also weighed. The dry matter (DM) weight of biomass (shoots and roots) was expressed in grams per plant (g plant$^{-1}$).

#### 2.3.2. Yields and Yield Components

Harvesting was performed at full maturity and was restricted to the four center rows per plot (3 m × 3 m). The final plant stands per plot were recorded. A subsample of 10 plants was randomly tagged in the harvest area (3 m × 3 m) of each plot before harvesting was carried out. Matured dry pods were hand-picked from the plants in the harvest area. The harvested pods were weighed and manually threshed, and grains were winnowed and weighed. Yields (pod and grain yield) per harvest area were converted to kg ha$^{-1}$. Later, the above-ground biomass (stover biomass) in the harvest area was cut and weighed, and a subsample (10 tagged plants) was collected, weighed, oven-dried in a forced-air oven (101-3AB Dry Oven Haunghua faithful Co. Ltd., Cangzhou, China) at 70 °C for 72 h and weighed again. The subsample (10 tagged plants) of dry weight measured per plot was used to adjust the stover yield per harvest area and then extrapolated to kg ha$^{-1}$.

#### 2.3.3. Pod Harvest and Harvest Index (PH and HI)

The pod harvest and harvest index (PH and HI) (Equations (1) and (2)) indicate the remobilization of photosynthates from the pod wall to the seed, and the remobilization of photosynthates or assimilates from the biomass to the seed yield, respectively.

$$\text{Pod harvest } [(\text{PHI})\%] = \frac{\text{Grain yield}}{\text{Pod yield}} \times 100 \qquad (1)$$

$$\text{Harvest index } [(\text{HI})\%] = \frac{\text{Grain yield}}{\text{Biomass yield}} \times 100 \qquad (2)$$

### 2.4. Economic Returns Analysis

The return on investment for adopting WV technology was estimated using the value:cost ratio (VCR) approach [22].

Profit (returns) is the difference between the total revenue and the total cost of production. Mathematically,

$$\text{Profit } (\pi) = \text{TR} - \text{TC} \tag{3}$$

where TR is the total revenue and TC is the total cost of production.

$$\text{TR} = \text{P} \times \text{Q} \tag{4}$$

where P and Q are the price (US\$) and quantity (kg) of cowpea produced, respectively using the technologies introduced.

$$\text{TC} = \text{TVC} + \text{TFC} \tag{5}$$

where TC is the total cost, TVC is the total variable cost and TFC is the total fixed cost. The TVC is the variable cost of inputs associated with each technology and estimated as the product of the price and the quantity of variable inputs used. These included the costs of seeds, ploughing, planting, weeding, WV fertilizer application, pest control, harvesting, post-harvest handling, insecticides, WV spray and transportation. The total fixed cost (TFC) was estimated using the depreciated value of fixed assets (inputs) such as hoes, knapsack sprayers, wellington boots, cutlasses, bicycle, protective clothing, etc. The straight-line depreciation method was used with a salvage value of zero. The TFC was kept constant across all technologies evaluated in this experiment. Since the experiment was conducted at a station, the output (yield) was adjusted 10% downwards to take care of the variation that would occur on a farm (farmers' fields). Hence, the yield was reduced by 10% in both 2021 and 2022, before the VCR was calculated.

The value–cost ratio (VCR) indicates the returns to the cost invested.

$$\text{VCR} = \frac{[\text{Profit}(\pi)]}{[\text{Total Cost}]} \tag{6}$$

Decision rule: if VCR > 1, it implies the technology is profitable; VCR < 1 implies the technology is not profitable and VCR = 1 implies that the technology breaks even. The decision rule is that a project (technology or products being used) is worth undertaking or recommending when the VCR is greater than one and vice versa. However, the project will break even when the VCR is equal to one.

### 2.5. Analyses of Soil Parameters

Soil chemical, biological and biochemical analyses were performed on samples collected in 2021.

#### 2.5.1. Chemical Analysis

Soil pH was measured using a Corning Model 320 pH meter after equilibrating 2 g of air-dried soil in 10 mL deionized water (soil/water ratio of 1:5) for 30 min. Soil available P was determined using the Bray-P1 method [23], and analysis was performed in duplicates. Soil ammonium ($NH_4^+$-N) and nitrate ($NO_3^-$-N) were determined using the colorimetric method [24]. Briefly, five grams of soil was extracted with 25 mL of 1 M potassium chloride (KCl). The results were expressed as mg kg$^{-1}$ soil. Soil available N was calculated as the sum of both soil $NH_4^+$-N and $NO_3^-$-N. All measurements were performed in duplicates.

#### 2.5.2. Soil Biological Analysis
Microbial Biomass Nitrogen (MBN)

Soil MBN was determined for a 15 g field-moist (oven-dried equivalent) soil sample (sieved) using the chloroform–fumigation–incubation method using 1 M KCl as an

extractant [25]. In brief, available N ($NH_4^+$-N + $NO_3^-$-N) from the fumigated (10 days) and non-fumigated (control) soil was quantified using the colorimetric method after KCl extraction [26]. The non-fumigated control values were subtracted from the fumigated values. The MBN was calculated using a $k_{EC}$ factor of 0.54 [27]. Each sample had duplicate analyses, and results were expressed on a moisture-free basis. Soil moisture was determined after drying at 105 °C for 48 h.

Potentially Mineralizable Nitrogen (PMN)

Soil PMN was determined following the anaerobic procedure [24]. Briefly, duplicates of five grams of soil were extracted with 25 mL of 1 M KCl after 7 days of pre-incubation at 60% water-filled pore space ($WFP_S$). One set of the soil was centrifuged using a Wincom 80-2C centrifuge (Wincom Company Ltd., Changsha, China) at 3000 rpm for 10 min after the 7 d pre-incubation period, and the other set of the soil was extracted with 25 mL of 1 M KCl after 30 days of incubation at 29 °C. The KCl filtrate collected before and after 30 days of incubation at 29 °C was filtered using filter paper (Whatman No. 42) and analyzed for ammonium ($NH_4^+$-N) and nitrate ($NO_3^-$-N). The difference in N ($NH_4^+$-N + $NO_3^-$-N) concentration before and after 30 days of incubation is used as an indicator of N mineralization [24].

Soil Microbial Respiration and Mineralizable Carbon

Soil microbial respiration and potentially mineralizable C were assessed following the titrimetric method to quantify the evolved $CO_2$ [19,28,29]. Briefly, 40 g of the soil was weighed into a 250 mL mason jar, wetted to approximately 60% water-filled pore space ($WFP_S$) and pre-incubated for seven days [28]. After the pre-incubation period, the soil was ventilated, and the moisture content was adjusted to 60% WFPS. Two vials of 5 mL 1M NaOH and a vial with 10 mL water were placed on the soil in the mason jar to maintain a humidified environment. The jars were sealed with lids and then incubated at 30 °C. The alkali traps were changed and titrated on day(s) 1, 3, 6, 12 18, 24 and 30. Unreacted alkali in the NaOH traps was precipitated with 2 mL 1M $BaCl_2$ and back-titrated with 1 M HCl using a phenolphthalein indicator to quantify the respired or evolved $CO_2$-C [30]. After each sampling time, the soil moisture content was checked and adjusted to 60% WFPS, and the vials with 5 mL of 1 M NaOH solution were refilled with fresh NaOH and resealed with the lid. The evolved $CO_2$-C was expressed as mg $CO_2$-C/100 g soil, whereas the cumulative released and carbon-mineralized $CO_2$-C were calculated based on the amount of $CO_2$–C released during different intervals of time in each treatment [31]. All determinations were made in duplicates and expressed on a dry weight basis.

Permanganate-Oxidizable Carbon (POXC)

Permanganate-oxidizable carbon (POXC) or ("active soil carbon") was analyzed based on [19,32,33]. Briefly, 2.5 g of air-dried soil was weighed into test tubes. To each test tube, 18 mL of deionized water and 2 mL of 0.2 M $KMnO_4$ stock solution were added, and tubes were shaken vigorously for 2 min. Tubes were allowed to settle for 10 min. After 10 min, 0.5 mL of the supernatant was transferred to a second 50 mL test tube and mixed with 49.5 mL deionized water. An aliquot (1000 μL) of each sample was transferred to a new glass vial. A set of internal standards, including a blank of deionized water, was prepared. A soil standard and a solution standard (laboratory reference samples) were added to serve as a check. All samples and checks were duplicated. Sample absorbance was measured using a GENESYS 30 spectrophotometer (Thermo Fisher Scientific, Waltham, MA, USA) at 550 nm. Permanganate-oxidizable C was determined using Equation (1) following [32]. POXC was calculated using Equation (7), which is based on [32]:

$$\text{POXC}\left(\text{mg kg}^{-1}\text{ soil}\right) = \left[0.02 \text{ mol L}^{-1} - (a + b \times \text{Abs})\right] \times \left(9000 \text{ mg C mol}^{-1}\right)\left(0.02 \text{ L solution Wt.}^{-1}\right) \qquad (7)$$

where 0.02 mol $L^{-1}$ was the concentration of the initial $KMnO_4$ solution, *a* was the intercept and *b* was the slope of the standard curve, Abs was the absorbance of the unknown soil sample, 9000 mg was the amount of C oxidized by 1 mol of $MnO_4$ changing from $Mn^{7+}$ to $Mn^{4+}$, 0.02 L was the volume of $KMnO_4$ solution reacted and Wt. was the mass of soil (kg) used in the reaction.

2.5.3. Soil Biochemical Analysis (Soil Enzyme Activities)

Dehydrogenase

Dehydrogenase enzyme activity was assayed following a colorimetric procedure [1,5], which was based on a colorimetric assay of 2,3,5-triphenylformazan (INTF) produced through a microorganism reduction of 2,3,5-triphenyltetrazolium chloride (TTC). The INTF was measured spectrophotometrically using a GENESYS 30 spectrophotometer (Thermo Fisher Scientific, Waltham, MA, USA) at 485 nm, and dehydrogenase activity was expressed as mg of INTF $kg^{-1}$ soil $h^{-1}$.

Alpha ($\alpha$)-Glucosidase

Alpha ($\alpha$)-glucosidase activity was assayed using a colorimetric method, using 4-nitrophenyl-$\alpha$-D-glucopyranoside as a substrate. Soil samples were incubated at 37 °C for 60 min and the reaction product *p*-nitrophenol was determined at 410 nm [5,34] using a GENESYS 30 spectrophotometer (Thermo Fisher Scientific, Waltham, MA, USA). Alpha ($\alpha$)-Glucosidase activity was expressed as mg of *p*-nitrophenol $kg^{-1}$ soil $h^{-1}$

Acidic Phosphatase

Acidic phosphatase activity determination was based on the hydrolysis of *p*-nitrophenyl phosphate added to the soil samples [35]. The phosphate releases *p*-nitrophenol, which was detected colorimetrically at 410 nm using a GENESYS 30 spectrophotometer (Thermo Fisher Scientific, Waltham, MA, USA). Acidic phosphatase activity was expressed as mg of *p*-nitrophenol $kg^{-1}$ soil $h^{-1}$.

Arylsulphatase

Arylsulphatase activity was determined using a colorimetric method, using *p*-nitrophenyl sulphate as a substrate. Soil samples were incubated at 37 °C for 1 h, and the reaction product (*p*-nitrophenol) was extracted using dilute alkali ($CaCl_2$ 0.5 M and NaOH 0.5 M) and determined at 410 nm [36] using a GENESYS 30 spectrophotometer (Thermo Fisher Scientific, Waltham, MA, USA), and activity of arylsulphatase was expressed as mg of *p*-nitrophenol $kg^{-1}$ soil $h^{-1}$.

Geometric Mean Enzyme Activity (GMea)

The GMea (a general index to integrate information from variables that possess different units and ranges of variation) of the assayed enzyme activities was calculated for each sample as follows [37]:

$$GMea = (\alpha - Glucosidase \times Acidic\ phosphatase \times Arylsulphatase \times Dehydrogenase\ activity)^{1/4} \tag{8}$$

*2.6. Statistical Data Analysis*

Data from all growth stages were pooled together, and the pooled mean data were subjected to the analysis of variance (ANOVA) using the Proc-mixed model in the SAS 9.4 statistical package (SAS Institute 2015). Significant differences between means were verified with Fisher's least significant test at 5% significance level. Pearson's correlation coefficient was used to evaluate the statistical correlation between the soil health indicators and the cowpea growth and yield parameters.

A principal component analysis (PCA) was performed for the variables using the SAS 9.4 statistical package (SAS Institute 2015). Principal components (PCs) refer to linear combinations of variables that account for the maximum variance in a data set. Principal

components having high Eigenvalues and variables with high factor loading best describe system attributes. Thus, only PCs with Eigenvalues $\geq$ 1.0 [38,39] were considered. Within each PC, only highly weighted factors (i.e., those with values $\geq$ 0.60) were maintained. The plant parameters used for PCA were the means of two years.

## 3. Results

### 3.1. Effect of Wood Vinegar Application on Biomass and Nodulation of Cowpea

The shoot biomass yield and nodule mass (nodule dry weight) of cowpea were significantly ($p < 0.05$) affected by WV application methods in 2021 and 2022. Wood vinegar's effect on shoot dry matter varied in both 2021 and 2022 (Table 2). In 2021, soil drenching (SD) and the foliar application (FA) of WV produced significantly greater ($p < 0.05$) shoot yield compared to the control (Con). However, in 2022, the foliar application of WV improved shoot yield compared to other treatments. Across the two seasons, soil drenching and the foliar application of WV produced more shoot yield compared to the control.

**Table 2.** Effect of wood vinegar application method on shoot dry matter, root dry matter and nodule mass of cowpea grown in Nyankpala, Ghana, during the 2021 and 2022 cropping seasons.

| Treatment | 2021 | 2022 | Mean (2 Years) | 2021 | 2022 | Mean (2 Years) | 2021 | 2022 | Mean (2 Years) |
|---|---|---|---|---|---|---|---|---|---|
| | Shoot Dry Matter (g plant$^{-1}$) | | | Root Dry Matter (g plant$^{-1}$) | | | Nodule Mass (mg plant$^{-1}$) | | |
| Control (Con) | 5.64 b | 5.51 b | 5.58 b | 0.51 a | 1.40 b | 0.95 b | 71.0 b | 30.6 c | 50.9 b |
| Foliar Application (FA) | 7.98 a | 7.40 a | 7. 69 a | 0.63 a | 1.89 a | 1.26 a | 81.1 a | 52.9 a | 67.0 a |
| Soil Drenching (SD) | 9.17 a | 5.56 b | 7.37 a | 0.60 a | 1.54 ab | 1.07 ab | 84.0 a | 42.2 b | 63.1 a |
| SEM | 0.517 | 0.395 | 0.321 | 0.051 | 0.140 | 0.072 | 2.54 | 1.170 | 1.317 |

Values shown are mean of four replicates ($n = 4$). Different lowercase letters within the same column indicate significant differences between treatments at $p < 0.05$ using Fisher's LSD. SEM is the standard error of mean.

Wood vinegar application's effects on root dry matter (root DM) was inconsistent in 2021 and 2022, respectively (Table 2) In 2021, root DM was not significantly ($p < 0.05$) affected by any WV application method, but in 2022, root DM was significantly ($p < 0.05$) influenced by WV application. The foliar application of WV produced significantly greater root DM compared to the control (Table 2). Wood vinegar application had a significant ($p < 0.05$) effect on nodule mass (Table 2). Irrespective of the WV application method, nodule mass improved significantly ($p < 0.05$) compared to the control. In 2021, soil drenching and the foliar application of WV produced superior nodule mass than the control, while in 2022, the foliar application of WV had the greatest nodule mass, followed by soil drenching and the control, which had the smallest nodule mass. The significant ($p < 0.05$) order for nodule mass was as follows: FA > SD > Con. Across the two seasons, foliar application and soil drenching were similar but significantly ($p < 0.05$) enhanced nodule mass compared to the control (Table 2)

### 3.2. Effect of Wood Vinegar Application on Yield and Yield components of Cowpea

The pod yield, grain yield and stover yield of cowpea were significantly ($p < 0.05$) affected by WV application methods in both 2021 and 2022. Pod yield was significantly ($p < 0.05$) influenced by WV application methods in both 2021 and 2022 (Table 3). Foliar application provided higher ($p < 0.05$) pod yields compared to the other treatments in 2021. Soil drenching also yielded significantly greater ($p < 0.05$) pods relative to the control. Pod yield order was as follows: FA > SD > Con. In 2022, soil drenching produced the highest ($p < 0.05$) pod yield, followed by foliar application, and the control had the least pod yield. The increased order for pod yield in 2022 ranked as follows: SD >FA > Con. Across the two seasons, greater pod yield was achieved with soil drenching, followed by foliar application and then by the control.

**Table 3.** Effect of wood vinegar application method on pod yield, grain yield and stover yield of cowpea grown in Nyankpala, Ghana, during the 2021 and 2022 cropping seasons.

| Treatment | 2021 | 2022 | Mean (2 Years) | 2021 | 2022 | Mean (2 Years) | 2021 | 2022 | Mean (2 Years) |
|---|---|---|---|---|---|---|---|---|---|
| | Pod Yield (kg ha$^{-1}$) | | | Grain Yield (kg ha$^{-1}$) | | | Stover Yield (kg ha$^{-1}$) | | |
| Control (Con) | 1356 c | 1228 c | 1292 c | 873 c | 854 c | 863 c | 2194 b | 4350 b | 3272 b |
| Foliar Application (FA) | 2139 a | 1364 b | 1752 b | 1120 b | 1114 b | 1117 b | 2804 a | 5805 a | 4303 a |
| Soil Drenching (SD) | 1910 b | 1835 a | 1873 a | 1265 a | 1416 a | 1341 a | 2357 b | 4328 b | 3343 b |
| SEM | 17.0 | 64.0 | 33.3 | 31.5 | 27.3 | 29.4 | 160.1 | 251 | 206 |

Values shown are mean of four replicates ($n = 4$). Different lowercase letters within the same column indicate significant differences between treatments at $p < 0.05$ using Fisher's LSD. SEM is the standard error of mean.

Soil drenching produced superior grain yield relative to the other treatments in both years (Table 3). Similarly, foliar application produced higher grain yield than the control in 2021 and 2022. The significant ($p < 0.05$) order for grain yield is as follows: SD > FA > Con. Increased grain yields due to WV application with respect to the control ranged from 45% with soil drenching to 28% with foliar application in 2021, and in 2022, 65% with soil drenching to 30% with foliar application. Hence, across the two seasons, soil drenching produced superior grain yield relative to the other treatments.

Stover yield was significantly ($p < 0.05$) affected by WV application methods (Table 3). Enhanced stover yield was produced via foliar application compared to the other application methods in both years (Table 3). Across the two seasons, foliar application enhanced stover yield relative to the control and soil drenching.

*3.3. Harvest and Pod Harvest Indices*

The harvest index was significantly influenced by WV application in both 2021 and 2022 (Figure 1). The application of WV through soil drenching resulted in a greater harvest index relative to foliar application and the control. Similarly, the control had a higher HI of cowpea compared with the foliar application of WV. In 2022, soil drenching and the foliar application of WV significantly increased the HI by 30% with respect to the control. The average (2 year) significant ($p < 0.05$) order for the HI is as follows: SD > FA = Con.

The pod harvest index indicates the partition of pod photosynthate to seed or grain yield. The pod harvest index showed a variable response in both years (Figure 1). In 2021, soil drenching with WV yielded a significantly greater PHI than the control and the foliar application of WV. In 2022, there was no statistical difference in PHI among the treatments applied; however, the trend suggests soil drenching and the foliar application of WV had higher additive effects than the control.

*3.4. Economic Returns on Wood Vinegar Application*

Returns on investment based on the value:cost ratio analysis revealed WV applied either through soil drenching or foliar application was more economically viable compared with the control. Soil drenching yielded higher returns compared to foliar application and the control in both 2021 and 2022 (Table 4; and Supplementary Table S2). Similarly, foliar application generated greater returns than the control in both years (Table 4). On average, across the two seasons (2 years), the order of net returns (profits) was SD (VCR = 2.24) > FA (VCR = 1.77) > Con (VCR = 1.27). The values indicate that for every U.S. dollar invested into cowpea production using WV applied via foliar application and soil drenching, the returns on investment were, on average, US$ 1.77 and US$ 2.24 for foliar application and soil drenching, respectively. For the control, US$ 1 invested returned US$ 1.27 (Table 4).

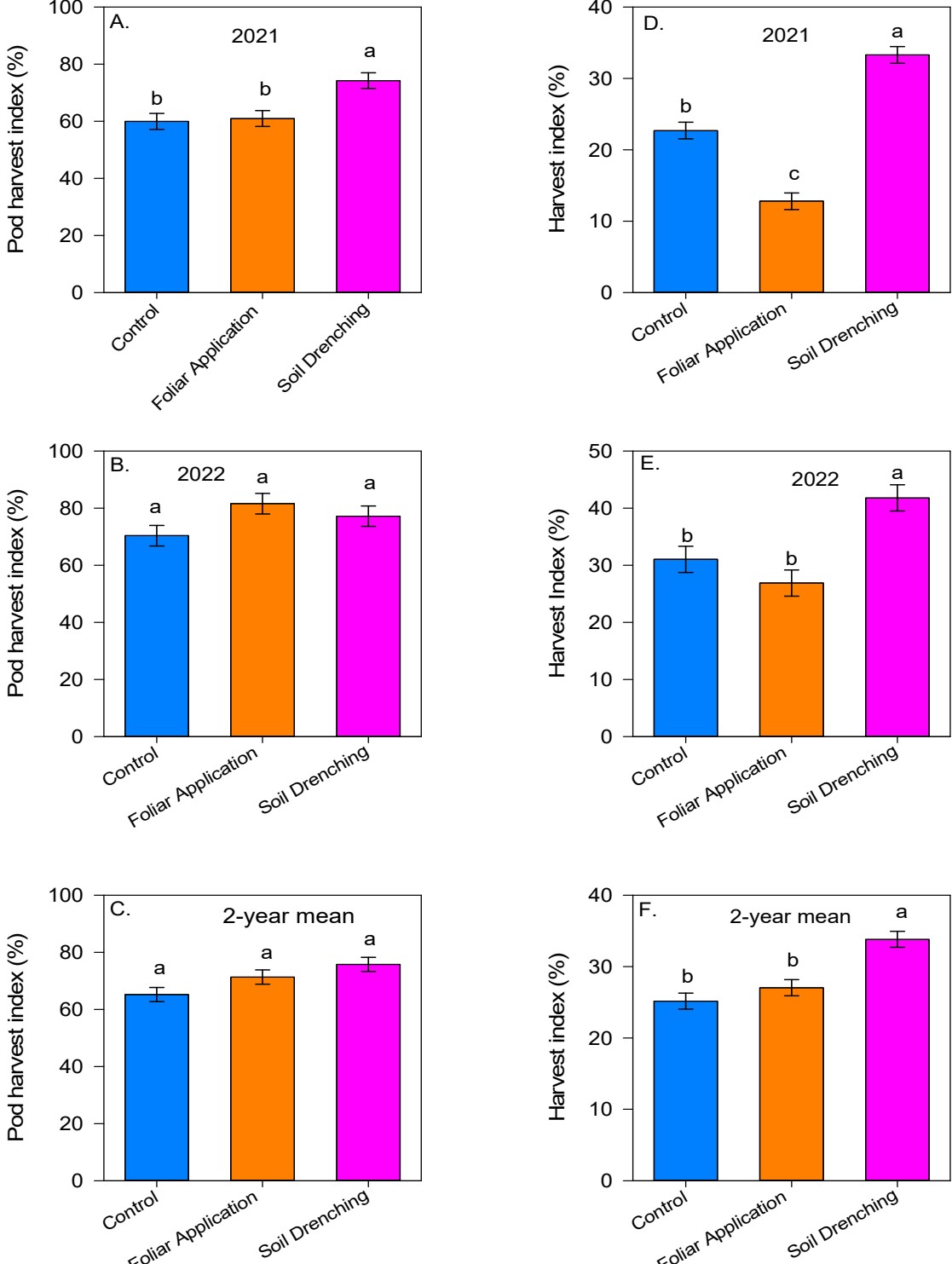

**Figure 1.** Effect of wood vinegar application method on pod harvest index (**A–C**) and harvest index (**D–F**) in Nyankpala, Ghana, in 2021 and 2022. Error bars represent standard error of mean (SEM) of four replicates (*n* = 4). Different lowercase letters indicate significant differences between treatments at *p* = 0.05 using Fisher's LSD.

**Table 4.** Benefit: cost analysis of wood vinegar application methods on the grain yield of cowpea grown in Nyankpala, Ghana, during the 2021 and 2022 cropping seasons. Values shown are the mean of four replicates (*n* = 4).

| Treatments | Adjusted Grain Yield | Percent (%) Increase | Gross Revenue | Variable Cost | Fixed Cost | Total Cost | Net Profit | Value: Cost Ratio (VCR) |
|---|---|---|---|---|---|---|---|---|
| Year: 2021 | kg ha$^{-1}$ | | | | US$ per hectare | | | |
| Control | 785.7 | | 978.01 | 286.31 | 138.31 | 424.62 | 553.43 | 1.30 |
| Foliar Application | 1008 | 28.3 | 1254.77 | 314.66 | 138.31 | 452.97 | 801.80 | 1.77 |
| Soil drenching | 1138.5 | 45.5 | 1417.22 | 326.76 | 138.31 | 465.08 | 952.14 | 2.05 |
| Year: 2022 | | | | | | | | |
| Control | 768.6 | | 1116.22 | 358.92 | 138.31 | 497.23 | 618.99 | 1.24 |
| Foliar Application | 1002.6 | 30 | 1456.06 | 387.28 | 138.31 | 525.59 | 930.47 | 1.77 |
| Soil drenching | 1274.4 | 66.3 | 1850.79 | 400.76 | 138.31 | 539.07 | 1311.72 | 2.43 |
| Mean (2-year) | | | | | | | | |
| Control | 776.7 | | 1047.42 | 322.61 | 138.31 | 460.93 | 586.49 | 1.27 |
| Foliar Application | 1005.3 | 29.4 | 1355.70 | 350.97 | 138.31 | 489.28 | 866.41 | 1.77 |
| Soil drenching | 1206.9 | 55.4 | 1627.56 | 363.76 | 138.31 | 502.07 | 1125.49 | 2.24 |

Control is the standard farmer practice.

### 3.5. Effect Wood Vinegar Application Methods on Soil Chemical and Microbial Properties

Soil pH and soil available phosphorus were not significantly ($p < 0.05$) affected by WV application (Figure 2). However, foliar application and soil drenching of WV slightly increased soil pH by 2.1% and 2.7%, respectively over the control (Figure 2A). Soil available phosphorus was also relatively higher after WV treatment relative to the control (Figure 2B). Soil available N ($NH_4^+$-N + $NO_3^-$-N) was significantly ($p < 0.05$) affected by both WV application methods (Figure 2C). The control recorded significantly ($p < 0.05$) greater available N compared to soil drenching and foliar application.

Potentially mineralizable N ($NH_4^+$-N + $NO_3^-$-N) (PMN) was significantly ($p < 0.05$) influenced by WV application methods (Figure 3). The control had greater PMN than foliar application and soil drenching. Soil drenching also improved PMN compared to foliar application.

Soil microbial biomass N (MBN) was significantly ($p < 0.05$) influenced by WV application methods (Figure 3). Soil drenching recorded the highest MBN, followed by foliar application and then the control. The significant ($p < 0.05$) order for increased MBN ranked as follows: SD > FA > Con (Figure 3). Soil permanganate-oxidizable carbon (soil POXC) differed significantly ($p < 0.05$) with application method (Figure 3). Soil drenching enhanced POXC compared to foliar application and the control.

Cumulative soil basal respiration (evolved $CO_2$) was not significantly ($p < 0.05$) affected by the type of WV application method, but the application of WV had an additive effect on cumulative soil basal respiration compared to no WV application (Figure 4A). Soil drenching had higher additive cumulative soil basal respiration relative to the other treatments (Figure 4A).

Soil mineralizable C differed significantly ($p < 0.05$) with WV application methods and incubation period. Soil drenching enhanced the greater mineralizable C pool relative to foliar application and the control (Figure 4C). Cumulative soil basal respiration and cumulative mineralizable C increased with a corresponding increase in incubation period (Figure 4B,D).

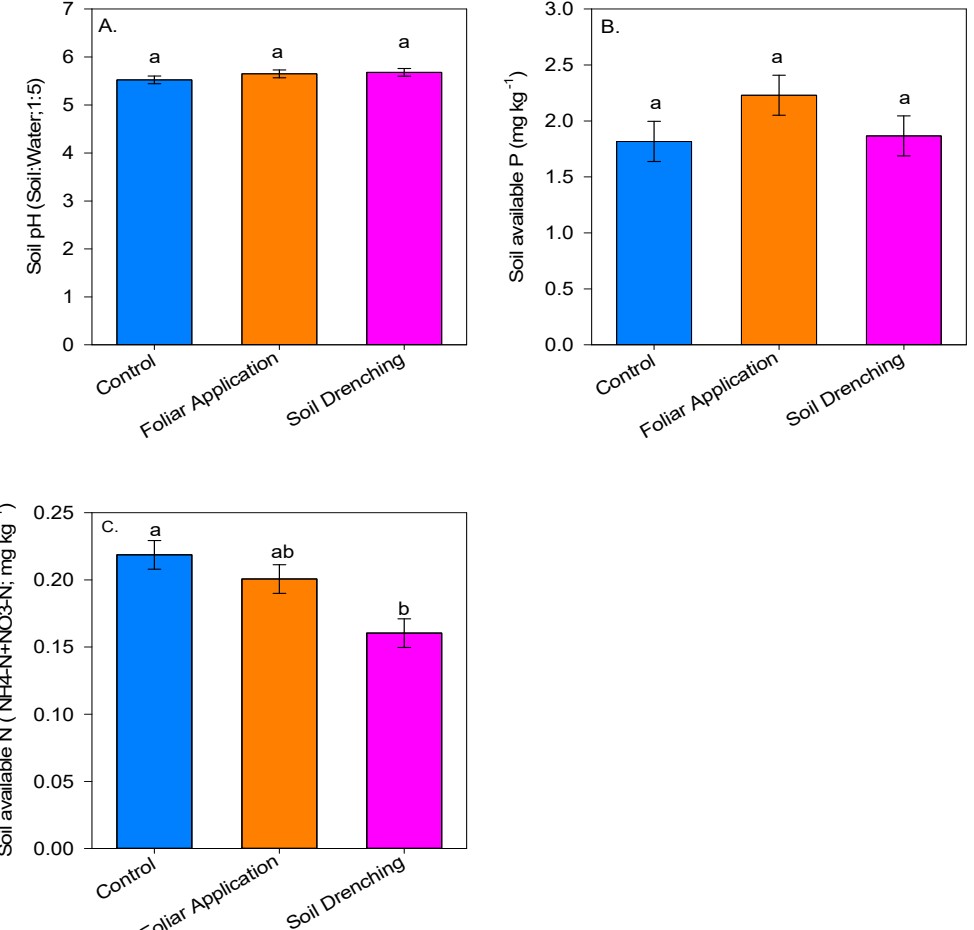

**Figure 2.** Effect of wood vinegar application methods on soil pH (**A**), soil available phosphorus (**B**) and soil available nitrogen ($NH_4^+$-N + $NO_3^-$-N) (**C**) in rhizosphere soil of cowpea grown in Nyankpala, Ghana, in 2021. Error bars represent SEM of four replicates (*n* = 4). Different lowercase letters indicate significant differences between treatments at *p* = 0.05 using Fisher's LSD.

### 3.6. Effect of Wood Vinegar Application Method on Biochemical Properties (Soil Enzymes Activities)

The activities of α-glucosidase, acid phosphatase, dehydrogenase and arylsulphatase were significantly (*p* < 0.05) affected by application method (Figure 5). Both soil drenching and the foliar application of WV produced similar and significantly greater (*p* < 0.05) activity of α-glucosidase, acid phosphatase and arylsulphatase than the control (Figure 5). For dehydrogenase activity, soil drenching had a significantly higher (*p* < 0.05) dehydrogenase activity compared with the control. However, no statistical (*p* < 0.05) difference was found in dehydrogenase activity between foliar application and soil drenching (Figure 5).

The geometric mean of soil enzymes activities (GMea) was significantly (*p* < 0.05) affected by both WV application methods (Figure 4). Both soil drenching and foliar application showed significantly greater GMea values compared with the control.

### 3.7. Principal Component Analysis
Relationships between Plant Parameters and Soil Health Indictors

A total of 21 variables consisting of plant and soil parameters were used in the principal component analysis (PCA) (Table 5). Five PCs had Eigenvalues > 1 and accounted for 92.34% of the variance in the data (Table 5; Figure 6). PC1 explained the highest variance (52.11%), followed by PC2 (21.55%), PC3 (1.710%), PC4 (1.159%) and PC5 (1.054%) (Table 5).

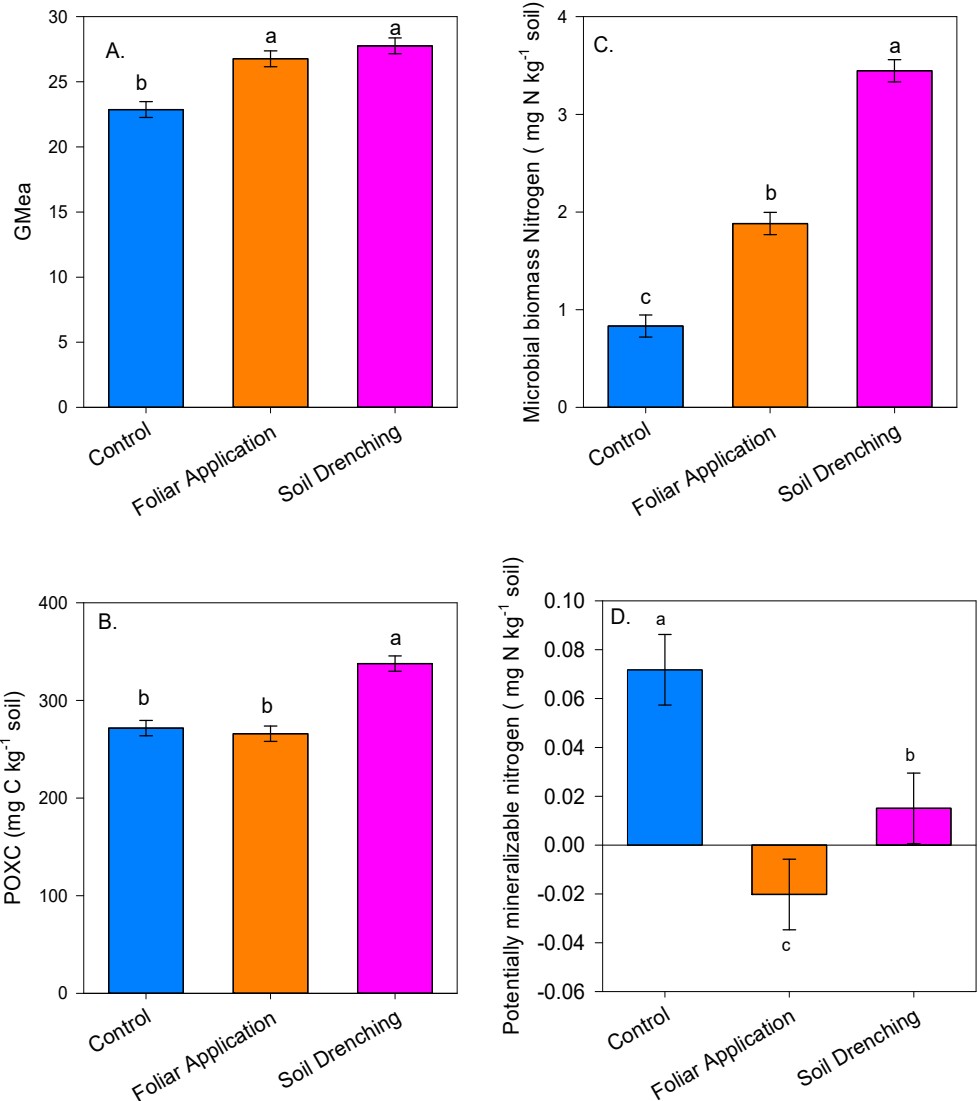

**Figure 3.** Effect of wood vinegar application methods on (**A**) geometric mean of enzymes activities (GMea) (**B**) permanganate-oxidizable C (POXC) (**C**) microbial biomass nitrogen (**D**) potentially mineralizable nitrogen (PMN) in rhizosphere soil of cowpea grown in Nyankpala, Ghana, in 2021. Error bars represent SEM (*n* = 4). Different lowercase letters indicate significant differences between treatments at *p* = 0.05 using Fisher's LSD.

The extracted PC1 can be regarded as the crop yield vector and had positive (+) and negative (−) factor loadings, which correlated positively and negatively with the axes, respectively (Table 5; Figure 6). The highly weighted factor loadings under PC1 included shoot DM, nodule mass, pod yield, grain yield, acid phosphatase, α-glucosidase, aryl-sulphatase, dehydrogenase, GMea, available phosphorus, available nitrogen, potentially mineralizable N, microbial biomass nitrogen, cumulative evolved $CO_2$, and cumulative mineralizable C. Hence, these variables were selected as the best data set for crop yield. Among the selected variables, it is only available nitrogen and potentially mineralizable N that had negatively weighted factor loadings in PC1 (Table 5; Figure 6).

PC2 can be regarded as the biomass production vector, and its highly weighted loading factors were the harvest index (HI), POXC and stover yield (Table 5; Figure 6). Among these variables, it is only stover yield which had a negatively weighted loading factor (Table 5; Figure 6).

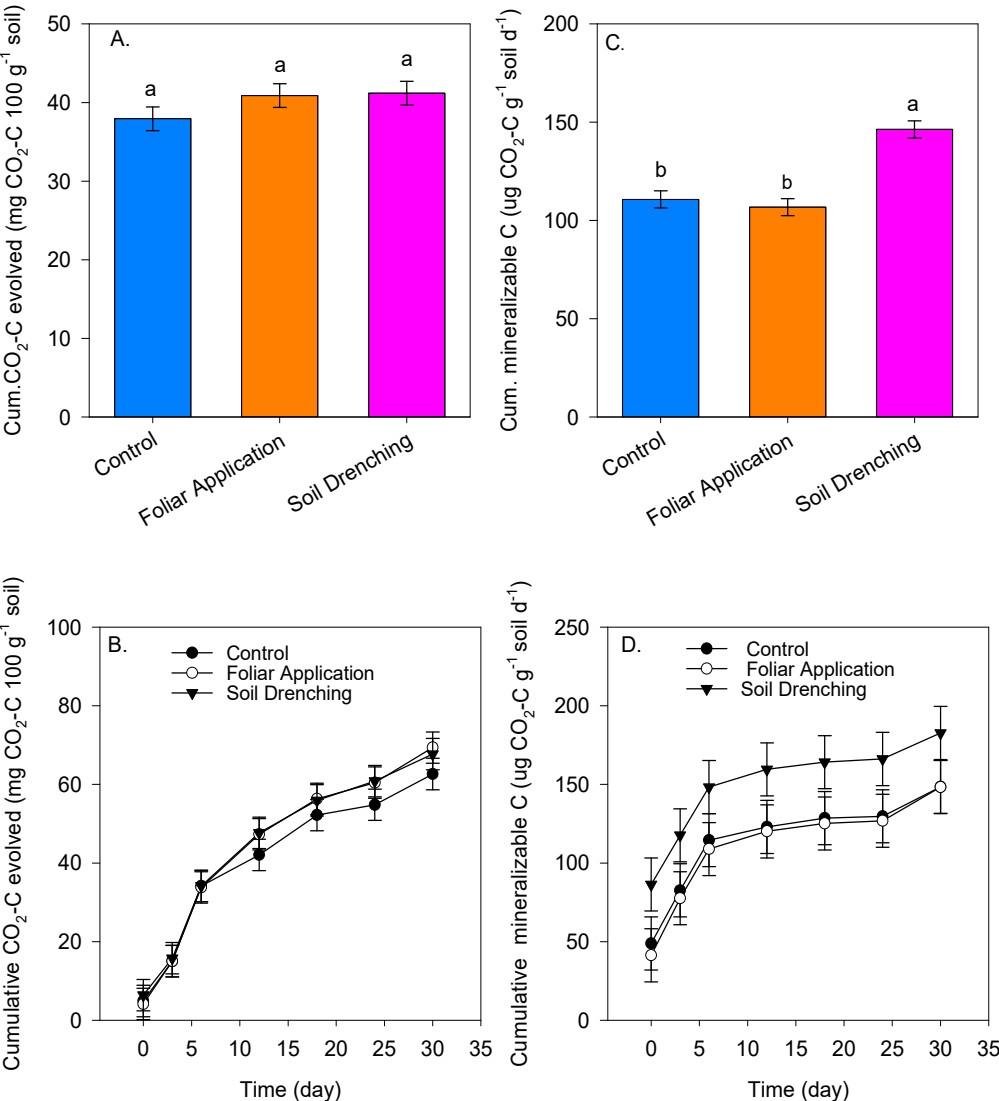

**Figure 4.** Effect of wood vinegar application methods on activity of (**A**) cumulative $CO_2$ evolved, (**B**) cumulative mineralizable C (POXC), (**C**) cumulative $CO_2$ evolved over the incubation time and (**D**) cumulative mineralizable cover the incubation period in rhizosphere soil of cowpea grown in Nyankpala, Ghana, in 2021. Error bars represent SEM (*n* = 4). Different lowercase letters indicate significant differences between treatments at *p* = 0.05 using Fisher's LSD.

The highly weighted variables in PC3 and PC4 were pod harvest index (PHI) and soil pH, respectively; hence, these variables were selected (Table 5). In PC5, the highly weighted variable was root DM, but the factor loading value was approximately 0.60 (Table 5).

Grain yield and pod yield are clustered together and were positively correlated with soil biochemical properties such as arylsulphatase, dehydrogenase, acid phosphatase, α-glucosidase, geometric mean enzyme activity (GMea), permanganate-oxidizable C (POXC) and microbial biomass nitrogen (MBN) (Table 5 and Table S3; Figure 6). In all these cases, the correlation was significant between soil biochemical properties, labile nutrients pool (POXC and MBN) and cowpea biomass production and yields (pod yield and grain yield) at *p* < 0.05.

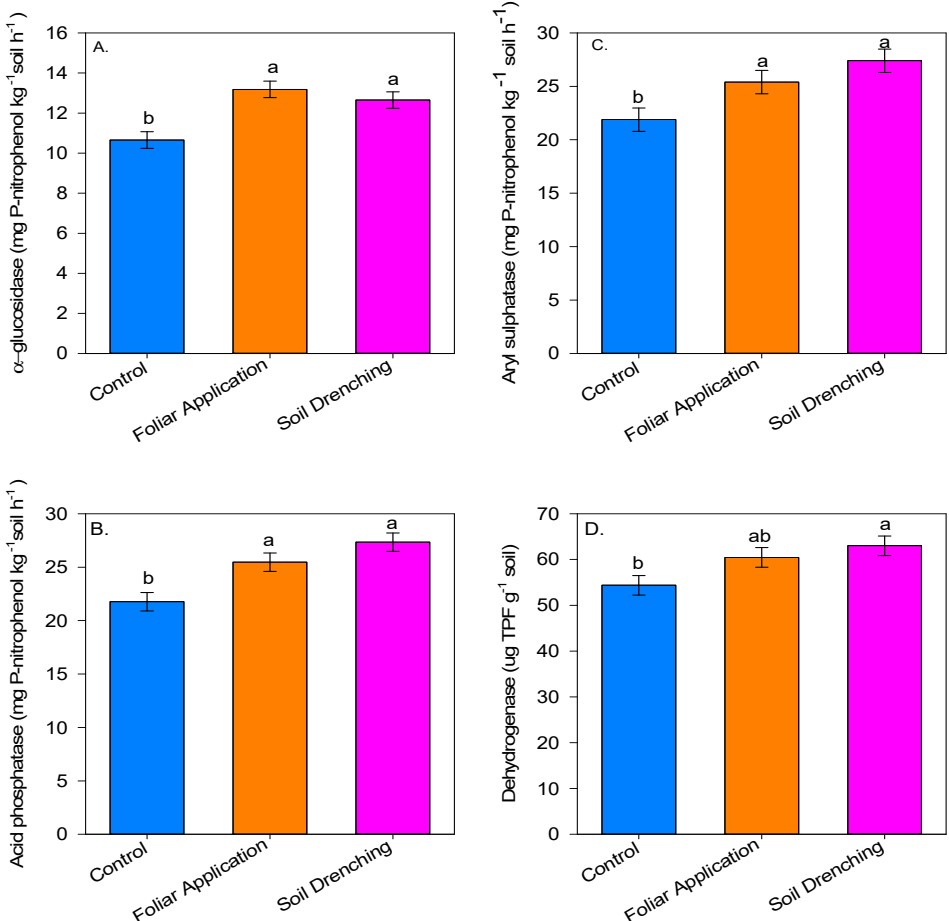

**Figure 5.** Effect of wood vinegar application methods on activity of (**A**) *α*-glucosidase, (**B**) acid phosphatase, (**C**) arylsulphatase and (**D**) dehydrogenase in rhizosphere soil of cowpea grown in Nyankpala, Ghana, in 2021. Error bars represent SEM (*n* = 4). Different lowercase letters indicate significant differences between treatments at *p* = 0.05 using Fisher's LSD.

**Table 5.** Principal component analysis (PCA) of cowpea productivity parameters and soil health Indicators.

| Statistical Parameter | PC 1 | PC 2 | PC 3 | PC 4 | PC5 |
|---|---|---|---|---|---|
| Eigenvalue | 10.944 | 4.526 | 1.710 | 1.159 | 1.054 |
| % of variance | 52.11 | 21.55 | 8.14 | 5.52 | 5.02 |
| Cumulative percent (%) | 52.11 | 73.67 | 81.81 | 87.33 | 92.34 |
| **Factor loading/Eigenvector** | | | | | |
| Parameters | PC 1 | PC 2 | PC 3 | PC 4 | PC5 |
| Shoot DM | **0.868** | −0.073 | 0.037 | 0.250 | −0.161 |
| Root DM | 0.514 | −0.554 | −0.215 | 0.168 | **0.561** |
| Nodule mass | **0.897** | −0.212 | −0.144 | 0.137 | 0.051 |
| Pod yield | **0.968** | 0.187 | 0.016 | −0.028 | −0.145 |
| Grain yield | **0.858** | 0.486 | 0.014 | 0.002 | −0.125 |
| Stover yield | 0.531 | **−0.805** | 0.087 | 0.120 | 0.091 |
| PHI | 0.554 | 0.172 | **0.661** | 0.217 | 0.057 |
| HI | 0.334 | **0.838** | 0.162 | −0.309 | −0.055 |
| Soil pH | 0.462 | 0.223 | −0.154 | **0.791** | 0.244 |
| Available phosphorus (P) | 0.445 | −0.469 | −0.402 | −0.320 | 0.369 |
| Available nitrogen (N) | **−0.803** | 0.350 | 0.230 | 0.182 | 0.265 |
| APHASE | **0.876** | 0.274 | −0.283 | −0.023 | 0.222 |

**Table 5.** *Cont.*

| Parameters | Factor loading/Eigenvector | | | | |
| --- | --- | --- | --- | --- | --- |
| | PC 1 | PC 2 | PC 3 | PC 4 | PC5 |
| ADASE | **0.865** | −0.213 | −0.031 | −0.138 | 0.215 |
| ASHASE | **0.830** | 0.285 | −0.368 | −0.019 | 0.241 |
| DHASE | **0.616** | 0.349 | 0.514 | −0.008 | 0.052 |
| GMea | **0.941** | 0.196 | −0.097 | −0.050 | 0.223 |
| POXC | 0.363 | **0.855** | −0.264 | −0.125 | −0.205 |
| Microbial biomass N | **0.762** | 0.548 | 0.083 | −0.017 | −0.201 |
| Potentially mineralizable N | **−0.760** | 0.482 | 0.049 | 0.082 | 0.290 |
| Cumulative evolved $CO_2$ | **0.697** | −0.593 | 0.355 | −0.052 | 0.062 |
| Cumulative mineralizable C | **0.669** | −0.387 | 0.487 | −0.289 | 0.132 |

Factor loadings are considered highly weighted when value(s) within each PC is/are $\geq$ 0.60 (60.0%). Factor loadings in bold numbers were considered highly weighted variables in each PC and were retained as the minimum data set (MDS). Shoot DM = Shoot dry matter; Root DM = Root dry matter; Nodule mass; Pod yield; Grain yield; Stover yield; PHI = Pod harvest index; HI = Harvest index; Soil pH; Available phosphorus; Available nitrogen ($NH_4^+$-N+$NO_3^-$-N); APHASE = acid phosphatase; ADASE = alpha-glucosidase; DHASE = dehydrogenase; ASHASE = arylsulphatase; GMea = geometric mean enzyme activity; POXC = permanganate-oxidizable C; Microbial biomass N; Potentially mineralizable N; Cumulative evolved $CO_2$; and Cumulative mineralizable C.

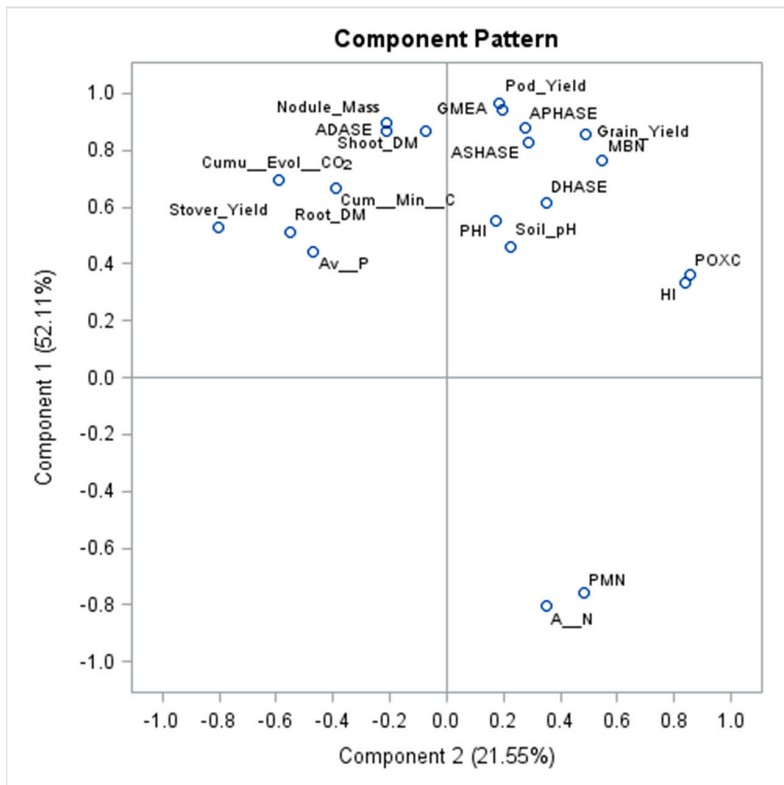

**Figure 6.** Principal component (PC) scores of all parameters (plants and soil) in the first two PCs. Percentage of total variance explained by each PC is indicated in brackets. Factor loadings are considered highly weighted when values within each PC are greater than 0.60 (>60.0%). Shoot_DM = Shoot dry matter; Root_DM = root dry matter; Nodule_Mass = nodule mass; Pod_Yield = pod yield; Grain_Yield = grain yield; Stover_Yield = stover yield; PHI = pod harvest index; HI = harvest index; Soil_pH = Soil pH; A_N = available N ($NH_4^+$-N+$NO_3^-$-N); Av_P = available phosphorus; APHASE = acid phosphatase; ADASE = alpha-glucosidase; ASHASE = arylsulphatase; DHASE = dehydrogenase; GMEA (GMea) = geometric mean enzyme activity; POXC = permanganate-oxidizable carbon (C); MBN = microbial biomass nitrogen; PMN = potentially mineralizable N; Cumu_Evolv_$CO_2$ = cumulative evolved $CO_2$; Cumu Min_C = cumulative mineralizable C.

Similarly, shoot DM significantly ($p < 0.0.5$) correlated positively with dry matter biomass (root DM and nodule mass), yields (pod yield and grain yield), soil biochemical properties (arylsulphatase, dehydrogenase, acid phosphatase, $\alpha$-glucosidase), GMea and other biological indicators such as POXC and MBN (Tables 5 and S3; Figure 6)

## 4. Discussion

Sustainable agriculture focuses on developing or introducing technologies that produce safe and healthy food without compromising the health of the soil and the environment. This is because healthy soil forms the foundation for eliminating food and nutrition insecurity and poverty. This study assessed the effect of different application methods of wood vinegar (WV) on crop productivity and soil health.

### 4.1. Effect of Wood Vinegar Application on Biomass Yield and Nodulation of Cowpea

Results showed a positive effect of WV application on biomass production (shoot and root dry matter) and nodulation of cowpea. The enhanced shoot yield and nodulation (nodule mass) associated with WV applied via soil drenching and foliar application suggest that WV can stimulate the production of more above-ground biomass and superior nodule mass. Previous work also confirmed that wood vinegar application enhanced the vegetative growth and accumulation of dry matter in groundnut [17]. Similarly, Praveena et al. [15] reported that either the sole application of wood vinegar or the combined application of wood vinegar and inorganic fertilizer resulted in enhanced dry shoot matter (DM) production and growth (plant height, leaf area index and number of branches per plants) of green gram in a sodic soil. Since cowpea is a grain legume, and both groundnut and green gram are also grain legumes, we may infer that the application of PA products such as WV can stimulate shoot dry matter accumulation in grain legumes.

Although WV application effects on root dry matter were not consistent in both years, the results demonstrated that WV application had an additive effect on root biomass production, with the foliar application of WV showing greater superiority in stimulating root biomass production in comparison to the control. Perhaps the WV stimulated the plant roots to discharge more exudates, which enhanced the plant microenvironment, subsequently resulting in the increased uptake of resources (nutrients and water) and the greater production of shoot and root dry matter. This finding corroborates that of Mungkunkamchao et al. [40], who observed that wood vinegar application encouraged plant roots to discharge more exudates, resulting in an improved microenvironment in plants, culminating in increased nutrient uptake and greater root dry matter production and roots development. Likewise, Lu et al. [41] found that the application of low concentrations of wood vinegar (0.33- and 0.50-mL $L^{-1}$) improved fresh root biomass, root vigor and root growth in wheat. Root systems provide mechanical support for the plant and enhance the uptake of resources (water and nutrients). Root vigor is an index used to assess the performance of the root in relation to its ability to uptake water and nutrients from the soil [37]. The root releases exudates, and several enzymes are involved in metabolic activities in relation to plant growth and development. Of the enzymes, dehydrogenase has been linked with increased photosynthesis and respiratory activities [37]; thus, we speculate that the improved dehydrogenase associated with WV treatments could be attributed to enhanced root vigor, although root vigor was not directly measured in this study. Lu et al. [41] reported that increased dehydrogenase activity can be an indicator of improved root vigor because dehydrogenase is one of the important enzymes involved in photosynthesis and respiratory action.

### 4.2. Effect of Wood Vinegar Application method on Yield and Yield Component of Cowpea

This study's results demonstrated that WV is capable of improving the yield and yield components of cowpea. Wood vinegar applied as soil drench and foliar increased grain yield of cowpea by 45.5% and 28.3%, respectively over the control in 2021, and in 2022, by 66.3% and 30% with respect to the control. The enhanced pod yield and grain yield



of cowpea observed with WV applied via soil drenching and foliar spray corroborate the findings of Travero and Mihara [16], who observed that wood vinegar application increased the grain yield of soybean by 92% in comparison with the control. In addition, a study by Praveena et al. [15] affirmed that pyroligneous acid (PA) is beneficial in improving the yield of green gram in a sodic soil. Similarly, WV application improved the yield and seed nutritional quality of chickpea [42]. Since cowpea, green gram, soybean, and chickpea are legumes, it can be inferred that WV or PA may be important in enhancing the productivity of legume crops.

Among the two WV application methods investigated, soil drenching had a greater advantage in improving shoot dry matter, pod yield and grain yield compared to foliar application. Grain yield from soil-drenched WV was ~ 13% and 27% more than grain yield from foliar-applied WV in 2021 and 2022, respectively. The superior performance of soil drenching can be attributed to the mode of WV application. During soil drenching, there is a high possibility that WV was directly transported to the root zones, resulting in immediate assimilation by the roots, whereas in foliar application, the WV applied to the leaves of the plants was assimilated through the leaves before reaching the root zone.

In addition, the high harvest index (HI) and pod harvest index (PHI) associated with soil drenching could imply that WV application increased the uptake of nutrients (e.g., N, P, K), which resulted in greater partition of dry matter (biomass) and pod yield into grain yield. Hence, the result demonstrates that soil drenching with WV had greater potential to alter the soil environment, leading to improved plant productivity, which is very consistent with our research hypothesis.

Although foliar application had greater pod yield than soil drenching, its low grain yield relative to soil drenching can be attributed to the low partition of pod photosynthates (low PHI) to grain yield. Likewise, the high stover yield associated with foliar application relative to soil drenching and the control can be attributed to low partition of biomass dry matter (low HI) into grain yield. Hence, the foliar application of WV would lead to a greater production of biomass at the expense of grain yield. This observation corroborates the findings of Jothityangkoon et al. [17], who found that the foliar application of wood vinegar (PA) on groundnut resulted in a substantial production of biomass yield in peanut in comparison to seed yield and shelling percentage. Therefore, it can be inferred that the foliar application of wood vinegar (pyroligneous acid) would stimulate the production of more biomass relative to soil drenching with WV in legumes such as groundnut and cowpea. Since most farmers in northern part of Ghana practice agropastoral farming, the improved biomass yield (stover yield) that remains on the fields after harvest can be used as fodder or feed for livestock, which is an additional benefit from the foliar application of WV. Thus, for producers in agropastoral systems, the foliar application of WV may be an option for consideration because it yields substantial biomass and, at the same time, produces superior grain yield than the control (no WV application).

In general, the substantial difference in stover yield in both years (2021 and 2022) could be attributed to a difference in weather patterns. The year 2022 appears to have had suitable weather conditions for cowpea production because the amount of rainfall received during the cowpea production period (August–October) was lower compared to 2021. Likewise, mean monthly temperatures in 2022 were relatively low compared to 2021.

### 4.3. Economic Returns of Wood Vinegar Application to Cowpea Production

The net returns of using WV on cowpea were computed using the VCR. A VCR threshold greater than one is considered to be attractive for farmers to adopt a new technology or innovation that relates to fertilizer use [22]. The VCR indicated that all the application methods are economically viable because their values are greater than one. On average (2 years), foliar application and soil drenching with WV had VCRs of 1.77 and 2.24, respectively, while the control had a VCR of 1.27, indicating that every 1 US$ invested into cowpea production technology with WV and applied as a foliar spray or soil drench yielded more profits compared to no WV application. The VCR also revealed that soil drenching is the

most profitable method to adopt because it yielded the greatest return on investment. Foliar application was the next and yielded a greater return on investment compared to no WV application. Since soil drenching produced higher profits than foliar application, it implies that farmers who adopt soil drenching with WV are more likely to gain greater net returns on investment than their counterparts who apply WV using foliar application. Hence, soil drenching seems to be the most economically viable technology for consideration for cowpea production.

Nonetheless, we recommend soil drenching and the foliar application of WV for cowpea farmers and other grain legumes farmers to adopt because both methods are profitable. In addition, this result could provide a basis for policy makers, governments and non-governmental organizations to consider and scale up the use of WV in low-external-input legume cropping systems through the provision of subsidies because it is less expensive, more environmentally friendly and more likely to provide more benefits to smallholder farmers in the northern part of Ghana.

### 4.4. Effect of Wood Vinegar Application Method on Chemical and Microbiological Properties

This study revealed that WV application can improve soil health biological and chemical indicators which corroborates previous studies that PA positively altered soil biological and chemical properties [5,43]. Enhanced soil enzyme activities, microbial biomass N, mineralizable C and POXC were observed in treatments that received WV compared to the controls. The high MBN associated with foliar application and soil drenching signifies that WV can significantly improve or enhance the labile N pool (labile microbial N pool) relative to the non-application of WV. In addition, WV applied via soil drenching enhanced the MBN pool more than WV applied as a foliar spray. The improved MBN linked with soil drenching suggests that soil drenching with WV can induce more labile N from microbial sources or pools. Perhaps the direct application of WV to the soil stimulated less temporal stress on the soil microbial communities which are responsible for nutrient cycling. The high MBN can also be attributed to the increased discharge of exudates from the roots as plants grow. Root exudates are important sources of substrates for microbes.

On the other hand, a decline was observed in PMN pools for treatments that received WV. The relatively low PMN pool associated with WV treatments suggests WV can partly inhibit or prevent the faster mineralization of organic N pools to readily available inorganic N pools, thus facilitating the storage of organic N in the soil (nitrogen sink). In addition, the gradual turnover of the organic N pool to a readily available inorganic N pool may have occurred at a stage that synchronized with peak N uptake by the cowpea, hence the improved shoot dry matter, nodulation grain yield and yield components observed with WV treatments compared to the control. The results also revealed that soil drenching of WV produced superior PMN relative to foliar application which implies that WV applied via soil drenching could contribute to more labile N pool (PMN). Perhaps, soil drenching of WV provided readily available substrate for the soil microbial communities in the rhizosphere to utilize for the metabolic activities including the mineralization or decomposition of soil organic matter (SOM) to release nutrients.

Permanganate-oxidizable carbon reflects a fraction of the labile soil C pool. The enhanced POXC ensuing from soil drenching with WV indicates that WV applied through soil drenching can potentially improve the labile carbon or active carbon pool of the soil. This may have resulted from the soil microbial communities utilizing the WV as a readily available substrate for their metabolic functions. Since POXC has been linked to soil microbial biomass and particulate organic C, which are fundamental for soil C cycling [44], soil drenching with WV could result in the better stabilization of POXC or labile C compared to the other methods. The increased stabilization of POXC or labile C has a long-term positive impact on soil organic C stabilization and its associated benefits for soil.

Although soil basal respiration was not significantly changed according to the type of WV application method, WV addition via soil drenching and foliar application slightly

increased soil basal respiration compared with the control. An increase in soil basal respiration due to WV addition suggests that WV had an additive effect of stimulating increased microbial activity or microbial respiration. The high cumulative soil mineralized C linked with soil drenching seems to support the assertion that WV can stimulate enhanced microbial activity or microbial respiration, thus accelerating soil organic matter decomposition. On the contrary, previous work has found that the application of wood vinegar did not accelerate the decomposition of native organic matter; thus, wood vinegar application helps store organic C in the soil [5], enabling the soil to serve as a carbon sink. The superior cumulative soil mineralizable C and POXC recorded with soil drenching with WV imply that WV applied via soil drenching was more efficient in enhancing microbial activity or microbial respiration, with subsequent positive effects on C mineralization and POXC pool. Thus, increased microbial activity is crucial in nutrient transformation and supply in the soil system.

Additionally, soil pH and available P were not significantly increased by means of WV application, but results revealed that WV addition could marginally increase soil pH and the available P pool. Becagli et al. [45] reported the application of WV did not alter the soil pH of *Vicia faba* var. *minor.* Similarly, Jeong et al. [46] observed that WV had no significant influence on the available N, P and K in the soil, although trends suggest WV could slightly improve the availability of these nutrients in the soil. This study suggests the possibility of WV slightly increasing soil pH and enhancing available P, confirming the previous work of Jeong et al. [46]. We observed that WV application inhibited or suppressed available N and PMN pools, suggesting that the application of WV may not enhance the available N and PMN pools. Thus, the application of WV may be optional to improve available N and PMN pools.

### 4.5. Effect of Wood Vinegar Application Method on Biochemical Properties

The high activity of acidic phosphatase, α-glucosidase, dehydrogenase and arylsulphatase resulting from WV being applied via foliar application and soil drenching suggests WV can potentially stimulate higher soil enzyme activities regardless of the application method used. Thus, the method of WV application has no significant effect on soil enzyme activities. On average, WV application increased acidic phosphatase, α-glucosidase, dehydrogenase and arylsulphatase by 65.9%, 63.9%, 66% and 66.0%, respectively, over the control. This finding corroborates with Becagli et al. [1] and Cardelli et al. [5,47], who found an increase in soil enzyme activities after the addition of wood vinegar. Thus, wood vinegar treatments generally increase enzyme activities as documented in previous works [6,45,48].

Additionally, acidic phosphatase and arylsulphatase are extracellular enzymes responsible for the cycling of phosphorus and sulphur, respectively, and their high abundance after WV application implies that WV can increase the metabolic activities of these microbes in the decomposition or mineralization of P and S. Alpha glucosidase is also an extracellular enzyme that is responsible for the cycling of carbon, and their high presence in soil treated with WV can be an indication that WV stimulated a high abundance of this microbe involved in the degradation of cellulose material. On the other hand, dehydrogenase is an intercellular enzyme, only available in viable cells and very sensitive to management (pollutants/heavy metals) [49]. Dehydrogenase facilitates the decomposition of lignin and thus improves C cycling through the release of nutrients from plant residues [50,51]. An increase in dehydrogenase activity may be linked to an increase in the metabolic function of the soil microbial population [49]. The high dehydrogenase associated with soil drenched with WV may imply that WV enhanced the efficiency of the soil microbial community structure.

Dehydrogenase activity is usually enhanced by the labile organic matter pool [52]; thus, the superior dehydrogenase activity associated with soil drenched with WV can be attributed to the high POXC and mineralizable C observed after WV was applied through soil drenching. XinCheng et al. [41] observed that wood vinegar application appears to stimulate cell growth and acts as a catalyst for microbe and enzyme activation. Since soil enzymes are responsible for the decomposition and recycling of nutrients from organic

residues in an ecosystem, enhanced soil enzyme activities are positive indicators of improved soil health, especially biological indicators. Thus, the improved arylsulphatase, acidic phosphatase, dehydrogenase and α-glucosidase activities associated with WV application through foliar spray and soil drenching is an indication that these two methods are probably the most efficient in improving the biological health of the soil. The improved enzyme activities may have contributed to the enhanced grain yield observed after the WV treatments. Wood vinegar application has been found to increase enzyme activities, creating a conducive soil environment for enhanced crop production [6].

The geometric mean of enzyme activities (GMea) is a good index for evaluating soil quality as it can be linked with other soil properties (physico-chemical properties) or other soil biological functions. It is an integrated soil enzyme index and provides early information on modification in soil quality due to management [53]. Since soil enzyme activities respond to rapid changes in management, they become useful indicators for assessing soil quality [54–57]. The importance of the GMea index as a soil quality indicator has been reported by Bastida et al. [58]. In the present study, foliar application and soil drenching had higher GMea values than the control. The high GMea values for foliar application and soil drenching indicate improved soil quality, whereas the low GMea values associated with the control (no WV application) imply a decline in soil quality. Previous studies have also proven that high GMea values and low GMea values correspond to improved soil quality and a loss in soil quality, respectively [1,54,55]. In effect, WV addition via soil drenching and foliar application seems to be beneficial for the soil microbial community, resulting in enhanced soil quality. Hence, the enhanced grain yield observed after the WV treatments can be attributed to improved soil quality.

*4.6. Relationship between Plant Parameters and Soil Health Indicators*

The PCA results suggest the crop yield (grain and pod yields) parameters were significantly influenced by soil biological activities, specifically soil enzymes, GMea and liable nutrient pools such as the labile C (POXC) and N (MBN) pools. Wood vinegar application probably enhanced soil biological function, which subsequently culminated in improved grain and pod yields. In addition, biomass production parameters (nodule mass, root DM and shoot DM) and stover yield were significantly influenced by both soil biological (basal respiration and the mineralization of organic matter and enzymes; α-glucosidase) and chemical (available P) functions. Hence, the clustering of these parameters suggests biomass production can be enhanced by enzymatic activity as well as soil biological and chemical properties.

Interestingly, we found that available N and potentially mineralizable N (PMN) had a negative linear relationship with biomass production and yield parameters as well as biochemical, biological and chemical properties in the PCA. The decrease in available N and PMN with the corresponding increase in the yield parameters of cowpea implies that greater uptake of the available N (available N and PMN) pool resulted in greater biomass production, culminating in an enhanced yield and yield component. The enhanced biomass production perhaps contributed to increased root exudation, leading to higher microbial activity in the rhizosphere. Thus, the significant and highly positive linear relationship observed with biomass production (shoot DM, root DM and nodule mass) and available phosphorus, α-glucosidase, cumulative evolved $CO_2$ and mineralizable C represents gross primary productivity. These variables can be selected as a minimum data set for evaluating gross primary productivity (biomass production) and soil health.

## 5. Conclusions

Soil drenching and foliar application are the most promising methods of applying WV to enhance the productivity of cowpea. Among these methods, soil drenching with WV produced the most outstanding grain yield, whereas the foliar application of WV had a more profound effect on biomass (stover) yield at harvest. Wood vinegar application generated higher economic returns, and soil drenching with WV yielded greater economic return

than foliar application. Thus, cowpea producers who apply WV through soil drenching and foliar application would get greater profits or returns on their investments.

Regarding soil health, soil drenching and the foliar application of WV positively affected soil biological and biochemical properties. Notable changes included improved soil enzyme activity (arylsulphatase, acidic phosphatase, $\alpha$-Glucosidase and GMea), POXC, MBN and mineralizable C. An improvement in these indicators reflects enhanced soil quality/soil health. The remarkable changes observed in the soil health indicators we assessed are very consistent with the test hypothesis that the addition of WV to the crop would enhance or improve the health of the soil. Hence, apart from the economic gain due to improved grain yield after WV application, producers who incorporate WV into their production activities through soil drenching and foliar application will equally improve the quality or health of their soils, leading to better environmental benefits. A significant and strong linear relationship exists between biomass, pod and grain yield and soil parameters. Therefore, achieving high productivity depends on the successful integration of soil biological, biochemical and chemical properties.

**Supplementary Materials:** The following supporting information can be downloaded at: https://www.mdpi.com/article/10.3390/agronomy13102497/s1, Table S1: Profitability for Cowpea Production at Different WV application—VCR. Table S2: Profitability for Cowpea Production at Different OFA application—VCR. Table S3: Pearson's Correlation Coefficients for on Cowpea Productivity Parameters and Soil Health Indicators.

**Author Contributions:** Conceptualization, E.K.A., O.A.A. and M.Y.; methodology, E.K.A., O.A.A. and M.Y.; software, E.K.A., S.S. and M.Y.; validation, E.K.A., P.A.Y.A., E.O.D., E.O., M.Y., S.S., A.F.N., A.F.S. and V.K.A.; formal analysis, E.K.A., S.S., A.F.N., A.F.S. and T.K.T.; resources, E.K.A., O.A.A. and M.Y.; data curation, S.S., A.F.N., A.F.S. and M.Y.; writing—original draft preparation, E.K.A., P.A.Y.A., E.O.D., E.O. and M.Y.; writing—review and editing, P.A.Y.A., E.O.D., E.O., M.Y., J.K.A., V.K.A. and T.K.T.; supervision, E.K.A., O.A.A., S.S. and A.F.N.; project administration, E.K.A.; funding acquisition, E.K.A., O.A.A., M.Y. and T.K.T. All authors have read and agreed to the published version of the manuscript.

**Funding:** This work was supported by HJA Africa and in part by the Bill & Melinda Gates Foundation [OPP1198373] through IITA and ICRISAT on the project "Accelerated Varietal Improvement and Seed Delivery of Legumes and Cereals in Africa (AVISA). The APC was funded by the Bill & Melinda Gates Foundation [OPP1198373].

**Data Availability Statement:** Not applicable.

**Acknowledgments:** We are grateful to the field technicians, especially Tibo Ibrahim, Stephen Duodu and Jessica K. Ohene at the soil microbiology laboratory of CSIR-Savanna Agricultural Research Institute (SARI) for assisting with data collection from the field.

**Conflicts of Interest:** The authors declare no conflict of interest.

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
