# Peer review of "Wood Vinegar Promotes Soil Health and the Productivity of Cowpea"

_agronomy, doi:10.3390/agronomy13102497_

Round 1
Reviewer 1 Report
I have had the pleasure of reviewing your manuscript titled " Impact of Organic Farming Aid on Selected Soil health Indicators and Cowpea Productivity." I must commend your comprehensive and robust approach to this pertinent issue in the realm of agricultural science.
The issue of Organic Farming is indeed a pressing one, and your research not only highlights the problem but provides a tangible and innovative solution. Your study design, especially the different methods of applying Wood Vinegar either through soil drenching or foliar application is appropriately rigorous, enabling a holistic understanding of the variables at play. Your focus on a wide array of measurements, including nodulation, shoot biomass, yield, value: cost ratio, soil enzymes, soil respiration, microbial biomass nitrogen (MBN), potassium permanganate oxidizable carbon (POXC), mineralizable C, soil pH, available nitrogen and phosphorus is highly commendable. However, there are some crucial amendments required as follow:
Point 1: The term "Organic farming aid" (OFA) frequently used within the manuscript could potentially be deceptive, as it encompasses various Good Agricultural Practices used in organic farming. Consequently, I propose using a more accurate description aligned with the actual intervention utilized in your research, by which is "Wood Vinegar." instead of using the term OFA throughout the whole MS.
Point 2: The suggested title of the MS should represent accurately the performed experiments without unnecessary information. I would suggest the title to be “Wood Vinegar as a Pyroligneous Acid Promotes Soil Health Indicators and Productivity of Cowpea.”
Point 3: My next comment pertains to the introduction section. While the introduction is informative, there is an apparent lengthiness that might be addressed. Some sections i.e., Line 106-122 contain repetitive information. Condensing these portions could enhance the overall clarity and focus of the introduction.
Point 4: My next comment pertains to the treatments of your study as illustrated in line 157-160. Please correct the description of your treatments according to my recommendation in Point 1 as well as in the other sections of the M&M.
Point 5: My next comment pertains to the largely descriptive nature of your results. While the data you present offer valuable insights, there could be additional analyses to deepen the interpretation and implications of your findings. Specifically, I recommend complementing your trait-by-trait analyses with correlation analysis and Principal Component Analysis (PCA).
Point 6: My next comment pertains to the error values in table 1 and 2, it's crucial to address the need for error values in your data tables. Providing error values alongside the mean values in your tables is necessary to give an accurate representation of your data's variability and precision. The inclusion of error values, such as standard deviations, standard errors, or confidence intervals, helps to convey the spread and reliability of your data.
Point 7: In your manuscript, I noticed that further information could be beneficial in the captions of your tables, particularly regarding the number of replicates used in your experiments.
Including information about the number of replicates (n) in your captions is important for several reasons. Firstly, it provides the reader with a clear understanding of the sample size, which can influence the interpretation of the data, especially when considering variability and statistical significance. Secondly, it provides context for the error bars (if present) and helps to validate the statistical analysis conducted. Lastly, it adds to the overall transparency and reproducibility of your research, both of which are critical aspects of scientific inquiry.
Point 8: Your manuscript would benefit from some minor revisions to ensure accuracy and consistency in your citations, references, and abbreviations. Here are the examples:
(1)
28. Franke, A.C.; Schulz, S.; Oyewole, B.D.; Bako, S. INCORPORATING SHORT-SEASON LEGUMES AND GREEN MANURE 828 CROPS INTO MAIZE-BASED SYSTEMS IN THE MOIST GUINEA SAVANNA OF WEST AFRICA. Exp. Agric. 2004, 40, 463–829 479, doi:10.1017/S001447970400211X.
48. Koc, I.; Öðün, E.; Namli, A.; Mendes, M.; Kutlu, E.; Yardim, E. THE EFFECTS OF WOOD VINEGAR ON SOME SOIL MICRO-872 ORGANISMS. Appl. Ecol. Environ. Res. 2019, 17, doi:10.15666/aeer/1702_24372447.
The title of this references wrote with capitalized letters !!
(2)
42. Crawford, E.W.; Kelly, V.A. EVALUATING MEASURES TO IMPROVE AGRICULTURAL INPUT USE.
The title of this reference wrote with capitalized letters !! and it needs more information to be added !!
Additionally, Cross-Reference Citations: Please verify that all the references in your bibliography have been cited within the body of the text and vice versa. Ensuring a match between the cited literature and the reference list will eliminate potential confusion for readers and maintain the integrity of your scholarly work.
Point 9: While your manuscript displays a strong command of the topic and presents compelling findings, I noticed some minor language issues and inconsistencies throughout the text. These could potentially hinder the clarity of your message and disrupt the reader's engagement with your work.
I noticed some minor language issues and inconsistencies throughout the text.
Author Response
Dear Reviewer:
Greetings. We appreciate your review comments and believe it will improve the quality of our manuscript. Please below are the feedback to your comments.
Point 1: The term "Organic farming aid" (OFA) frequently used within the manuscript could potentially be deceptive, as it encompasses various Good Agricultural Practices used in organic farming. Consequently, I propose using a more accurate description aligned with the actual intervention utilized in your research, by which is "Wood Vinegar." instead of using the term OFA throughout the whole MS.
Yes, your suggestion is well noted and taken. Although there is a slight difference in quality
Point 2: The suggested title of the MS should represent accurately the performed experiments without unnecessary information. I would suggest the title to be “Wood Vinegar as a Pyroligneous Acid Promotes Soil Health Indicators and Productivity of Cowpea.”
Please your suggestion has been taken and we are grateful. However, we would modify it
“Wood Vinegar Promotes Soil Health and Productivity of Cowpea””
Point 3: My next comment pertains to the introduction section. While the introduction is informative, there is an apparent lengthiness that might be addressed. Some sections i.e., Line 106-122 contain repetitive information. Condensing these portions could enhance the overall clarity and focus of the introduction.
This is a good suggestion. We have worked on the introduction
Point 4: My next comment pertains to the treatments of your study as illustrated in line 157-160. Please correct the description of your treatments according to my recommendation in Point 1 as well as in the other sections of the M&M.
We have worked on line 157-160 and highlight that portion.
Point 5: My next comment pertains to the largely descriptive nature of your results. While the data you present offer valuable insights, there could be additional analyses to deepen the interpretation and implications of your findings. Specifically, I recommend complementing your trait-by-trait analyses with correlation analysis and Principal Component Analysis (PCA).
We agree. May be correlation could be of help. However, we felt that manuscript is already lengthy with a lot of data. Therefore, we humbly suggest that we leave out the correlation or PCA analysis. Nevertheless, we would be very happy to include if it is necessary to help explain some of the variables.
Point 6: My next comment pertains to the error values in table 1 and 2, it's crucial to address the need for error values in your data tables. Providing error values alongside the mean values in your tables is necessary to give an accurate representation of your data's variability and precision. The inclusion of error values, such as standard deviations, standard errors, or confidence intervals, helps to convey the spread and reliability of your data.
We agree, please we have provided the standard errors of mean difference (SEM) for your consideration. We have very grateful, this has helps to correct some inconsistencies with the assigning of the letters to mean values.
Point 7: In your manuscript, I noticed that further information could be beneficial in the captions of your tables, particularly regarding the number of replicates used in your experiments.
Please we have provided this information on the tables in the manuscripts (replicates were four (n=4), we are grateful for this valuable comments. It was an oversight from our side.
Including information about the number of replicates (n) in your captions is important for several reasons. Firstly, it provides the reader with a clear understanding of the sample size, which can influence the interpretation of the data, especially when considering variability and statistical significance. Secondly, it provides context for the error bars (if present) and helps to validate the statistical analysis conducted. Lastly, it adds to the overall transparency and reproducibility of your research, both of which are critical aspects of scientific inquiry.
Point 8: Your manuscript would benefit from some minor revisions to ensure accuracy and consistency in your citations, references, and abbreviations. Here are the examples:
We have revised some section of the manuscripts and highlighted them. Thanks for this suggestion.
(1)
- Franke, A.C.; Schulz, S.; Oyewole, B.D.; Bako, S. INCORPORATING SHORT-SEASON LEGUMES AND GREEN MANURE 828 CROPS INTO MAIZE-BASED SYSTEMS IN THE MOIST GUINEA SAVANNA OF WEST AFRICA. Exp. Agric. 2004, 40, 463–829 479, doi:10.1017/S001447970400211X.
- Koc, I.; Öðün, E.; Namli, A.; Mendes, M.; Kutlu, E.; Yardim, E. THE EFFECTS OF WOOD VINEGAR ON SOME SOIL MICRO-872 ORGANISMS. Appl. Ecol. Environ. Res. 2019, 17, doi:10.15666/aeer/1702_24372447.
The title of these references wrote with capitalized letters!!
We have also worked on this inconsistencies in the manuscript with regards to citation.
(2)
- Crawford, E.W.; Kelly, V.A. EVALUATING MEASURES TO IMPROVE AGRICULTURAL INPUT USE.
The title of this reference wrote with capitalized letters !! And it needs more information to be added!!
We have also worked on this inconsistencies in the manuscript with regards to citation.
Additionally, Cross-Reference Citations: Please verify that all the references in your bibliography have been cited within the body of the text and vice versa. Ensuring a match between the cited literature and the reference list will eliminate potential confusion for readers and maintain the integrity of your scholarly work.
Point 9: While your manuscript displays a strong command of the topic and presents compelling findings, I noticed some minor language issues and inconsistencies throughout the text. These could potentially hinder the clarity of your message and disrupt the reader's engagement with your work.
We worked on some section of the manuscripts to enhance clarity. Thanks for this suggestion.
Comments on the Quality of English Language
I noticed some minor language issues and inconsistencies throughout the text.
Again, we are very grateful for your valuable review comments.

Reviewer 2 Report
Dear authors,
The reviewed manuscript shows a very important topic concerning impact of organic farming aid on selected soil health indicators and cowpea productivity. Organic farming aid (OFA) is a biostimulant and a bio-pesticide that contains pyroligneous acid. It is verry important to improve the soil health and crop sustainable production used OFA in agricultural production. The results of this study have certain theoretical and technical significance for improving soil health indicators and cowpea productivity.
Below I present my notes on the individual chapters of the work:
1. Other management conditions should be described in 2. Materials and Methods, e.g., ploughing, fertilization, weeding, Pest control…..
2. It is recommended to describe the root sampling method in detail in 2.3.1.
3. “The final plant stands per plot at harvest were recorded before plants were manually uprooted. A subsample of 10 plants was taken from the harvest area of each plot. Later, matured dry pods were hand-picked from the plants in the harvest area.” in 2.3.2. Yields and Yield components. Usually, the actual output is measured according to the single collection of the community, or set aside the test area. What was the test area? If it was calculated by measure 10 plants. Was it the theoretical yield?
4. Whether the calculation formula is complete about Pod harvest and harvest index (PH and HI)? Is there %?
5. The “X” should be “×” in “PO X C”. It is appeared in parts of the manuscript.
6. Suggest to modify the GH₵ per hectare to US$ ha-1 in Table 3, and conversion the table datas.
7. Suggestion authors modify the NH4-N and NO3-N to NH4+-N and NO3--N in Figure 2, and it is appeared in multiple parts of the manuscript.
8. If “at P = .05 using Fisher LSD” in note of many tables is “at p < 0.05 using Fisher LSD”.
9. Most references are too old.

Author Response
Dear Reviewer:
Greetings. We appreciate your review comments, and it will contribute to the improve the quality of our manuscript. Please below are the feedback to your comments.
Below I present my notes on the individual chapters of the work:
- Other management conditions should be described in Materials and Methods, e.g., ploughing, fertilization, weeding, Pest control
This is a good observation; we have included this information in the manuscript. We are grateful for this suggestion.
- It is recommended to describe the root sampling method in detailin 2.3.1.
We have provided information on the root sampling method in the revised manuscript. We are thankful for this suggestion.
- “The final plant stands per plot at harvest were recorded before plants were manually uprooted. A subsample of 10 plants was taken from the harvest area of each plot. Later, matured dry pods were hand-picked from the plants in the harvest area.” in 2.3.2. Yields and Yield components. Usually, the actual output is measured according to the single collection of the community or set aside the test area. What was the test area? If it was calculated by measure 10 plants. Was it the theoretical yield?
We are sorry about the lack of clarity for the sampling methods on yield and yield component section of the manuscript. We revised that section.
Briefly, we actually, tagged 10 plants in the harvest area prior to harvesting of the matured pods. After harvesting of the matured pods, the above ground-biomass (stover yield) was cut and weighed. The tagged plants were taken as the sub-samples plant, weighed, oven dried for 48 h and weighed again. The dry weight of sub-samples plants was used to adjust the weights of the stover yield per harvest area before extrapolating to kg ha-1.
Please many thanks for asking and pointing this limitation. We are very grateful. We have also revised that portions of the write-up to improve clarity.
- Whether the calculation formula is complete about Pod harvest and harvest index (PH and HI)? Is there %?
Yes, we have added it to the formulae or equation. Thank you very much.
- The “X” should be “×” in “PO X C”. It is appeared in parts of the
Please, I think POXC is also acceptable too. Technically speaking POXC is the most acceptable form. We have harmonized all to POXC to reflect what we intend to write.
However, we can change it to POxC to fit the style of the journal. We are thankful for this suggestion.
- Suggest to modify the GH₵ per hectare to US$ ha-1in Table 3, and conversion the table datas.
We have revised table 3 and converted the values to US$ ha-1. Thank you
- Suggestion authors modify the NH4-N and NO3-N to NH4+-N and NO3--N in Figure 2, and it is appeared in multiple parts of the manuscript.
We have changed NH4-N and NO3-N to NH4+-N and NO3—N in the all MS. We are grateful for this observation and suggestion
- If “at P = .05 using Fisher LSD” in note of many tables is “at p < 0.05 using Fisher LSD”.
We have corrected P=0.05 to p < 0.05 to ensure consistency. We are grateful for this suggestion
- Most references are too old.
We have updated some citations especially those in the Material and Methods.
Again, many thanks for reviewing our manuscript.

Reviewer 3 Report
Dear Authors,
Please find my comments in the attached file.

Author Response
Response for Reviewer THREE
Manuscript ID: agronomy-2564761 Type: Article
Dear Reviewer:
Greetings. We are thankful for your review comments and believe it will improve the quality of our manuscript. Please below are the feedback to your comments.
Evaluation: Major Revision
- It is not recommended to use the same words or phrases as key words that appeared in the title of the manuscript. Therefore, the keywords (e.g. organic farming aid, soil health) that are included in the title should be deleted.
Please this is well noted, we have changed it. We are grateful.
- The Introduction should be improved. The cited publications are mostly relevant, but a significant number of the references cited are not from the latest 5 years. Authors need to refer to recent articles.
The introduction has been revised and some literature citations have been changed.
- The current state of the research field is not clearly highlighted.
The field has been rotated to maize and soybean.
- The Material and Methods chapter also needs clarification in several points. Weather conditions paragraph is missing.
The materials and methods section have been revised with more information added to enhance clarity. We are very grateful
- The authors mention only the average temperature and precipitation data for the experimental area.
Well noted. Information has been provided. We are grateful.
- I also propose to summarize the nutrient characteristics of soil in a table, as this would be more comprehensible.
Please we are grateful for this suggestion. We have the nutrient characteristics of baseline soil in a table.
- Line 165-166 was this amount applied per plot? What is the dose per hectare?
Please, No, we use 45ml of OFA/ wood vinegar (WV) in 22.5 L of tap water was applied to per treatment which consisted of four 4 m x 3 m. Technically, each plot received 11.25 ml of WV in 5.63 L of tap water. The doses per ha is (300 ml ha-1). Thank you for asking.
- Line 172-173 how many treatments were there in total? How was the frequency of treatment determined?
We have three treatments of WV consisting of a control, soil drenching and foliar application, replicated four times. The spraying was done on weekly basis starting at the R4 stage (4th leaf stage). We are grateful for asking
- Were other nutrients added? How much? What plant protection treatments were used?
Please no, we did not provide any blanket nutrients other than the wood vinegar.
- Plant protection such as weed, and pest controls were carried out.
We have provided that information into the manuscripts and highlighted it purple. Thanks for drawing our attention to it.
“Weeds were controlled manually with hoes and when necessary. Insects were controlled using K-Optimal (Lambda Cyhalothrine 15 g/L +Acetamipride 20 g/L EC) insecticide at 30 ml in 15 L of water and applied with a knapsack sprayer at an application rate of 300 ml ha-1. Chemical control of insects began at full flower to early pod stage (R5 stage) and was done weekly for 3 times. Later, the spraying frequency increased to every three (3) days to prevent insect infestation of the pods. However, the insecticide application stopped at the full pod maturity stage. In all six spraying regimes were done “
- In the Results chapter, I suggest evaluating the effect of different forecrop. There is a significant difference between the maize (2021) and fallow (2022) forecrop, and therefore it affects the results obtained.
Please it is well noted, sincerely, we did not anticipate seeing any differences, since results from the baseline soil analysis, or baseline nutrient characterization were still below the critical level (i.e. very low) required for crop production. Nonetheless, we admit that it could be limitation in this study which we can investigate in the future. Thanks for providing us with this suggestion.
- 39 citations out of 58 are referred to Introduction and Material and Methods. Only 19 new and some cited also in an introduction appeared in the Discussion. The balance should be slightly different.
Please it is well noted. Sincerely, it is the materials and methods that contain the old citations because we methodology that were developed long-term and are still being us. Therefore, we cited the original research articles who developed the method we used for them to get the credit. The old citations were mostly on the soil enzymes, soil chemical and biological properties.
In addition, it was pretty difficult to find more recent research on the topic being investigated in the context of the tropical climate and crop production especially legumes production. Therefore, we used those articles that we could easily and readily access. Nonetheless, we have provided some updated literatures.
Please we would be very grateful if could kindly connect us to any sources where we can get more recent publication related to the topic to support our research.
Again, we are very grateful for your suggestion.
- I suggest to further develop the Discussion. This chapter is partly a reiteration of the results described earlier and is therefore too long, and not the actual discussion of them.
Please we have revised the discussion section of the manuscript. This is a valuable comment that has help to fine tune the manuscript.
- I recommend the authors to compare their findings with the latest scientific results. Line 729
The is well noted but it was pretty difficult to find more recent research on the topic being investigated in the context of the tropical legumes production Therefore, we used those articles that we could easily and readily access. Please kindly connect us to any sources where we can get more recent publication related to the topic to support our research finding.
Again, many thanks for your useful comments, and spending time to review our paper.
- Please delete: the most . You cannot say that these are the most effective methods, only that they are effective methods, because you have not tested other application methods.
Please it is well noted. We have revised that conclusion.
- Line 738 Please correct oil to soil.
Many thanks for catching the typo, we have worked on it.
- References should be prepared strictly according to the Guides for Authors. There are some editorial mistakes, please check and correct all of them.
Please it is noted, many thanks for this suggestion, we have worked on it.
Again, we are very grateful for your review comments.

Reviewer 4 Report
The manuscript entitled “Impact of Organic Farming Aid on Selected Soil health Indicators and Cowpea Productivity” investigated the effects of different applied methods of organic farming aid on soil health and Cowpea performance with a two-year experiment. The work is interesting. It can be seen that the authors did a lot of works in the study.
Why did the title emphasize “Selected Soil health Indicators” rather than “soil” or “soil health”?
Line 24 and Line 285: What is POXC? potassium permanganate oxidizable carbon or permanganate-oxidizable carbon? Please clarify and be unitive.
Line 23: Will it be better if use “value-cost ratio”
The full name only needed to be mentioned when the abbreviation first used.
Line 123: it should be “hypothesized”
Line 133-134: Why are the longitude and latitude different in two years? Moreover, the longitude and latitude should be mentioned with east, west, north and south.
How did authors acquire OFA? The detail chemical component and manufacturer also should be provided.
Line 167-173: When and what stage of cowpea did the OFA applied?
Table 1 and 2: n=? the standard error should be added.
Line 496: “mode of” should be deleted.
The soil chemical and microbial properties only demined in 2021?
Line 580: “was” should be “were”
Line 729: it should be “The results of this study” or just “This study”
The conclusion is too long and should be only one paragraph as far as I am concerned.
Authors should carefully check throughout the manuscript for grammar and writing.
Authors should carefully check throughout the manuscript for grammar and writing.
Author Response
Response for Reviewer four
Dear Reviewer:
Greetings. We are thankful your review comments. It will help improve the quality of our manuscript. Please below are the feedback to your comments.
- Why did the title emphasize “Selected Soil health Indicators” rather than “soil” or “soil health”?
The focus was assessing the indicators of soil health but after careful deliberation and suggestions from other reviewer, we have changed the topics to Wood Vinegar promotes Soil Health and Cowpea Productivity. We are grateful for your suggestion.
- Line 24 and Line 285: What is POXC? potassium permanganate oxidizable carbon or permanganate-oxidizable carbon? Please clarify and be unitive.
We are sorry, it is an oversight, the correct terminology should be permanganate-oxidizable carbon. Thanks for catching the oversight.
- Line 23: Will it be better if use “value-cost ratio”
Please it is well noted, we have done the correction
- The full name only needed to be mentioned when the abbreviation first used.
Please it is well noted. Correction has been done.
- Line 123: it should be “hypothesized”
Please it is well noted. The correction has done
- Line 133-134: Why are the longitude and latitude different in two years? Moreover, the longitude and latitude should be mentioned with east, west, north and south.
Please it is well noted. But the fields were quiet closed to each other.
- How did authors acquire OFA? The detail chemical component and manufacturer also should be provided.
OFA was soured from commercial producer and vendor called HJA Africa. Information on the chemical composition has been provided in the revised manuscript.
- Line 167-173: When and what stage of cowpea did the OFA applied?
Please we applied the OFA at V4 stage (4-leaf stage which 25 days after planting)
- Table 1 and 2: n=? the standard error should be added.
We have provided the standard and the number of replicates, we are grateful for this suggestion
- Line 496: “mode of” should be deleted.
Please it is well noted.
- The soil chemical and microbial properties only demined in 2021?
Yes, the fields were near to each other, and we did not anticipate significantly in the microbial and soil chemical properties. Nonetheless, we do admit, it is a limitation in our study. In the future, we will consider two-year analysis to overcome this limitation.
We are very grateful for bringing this to our attention.
- Line 580: “was” should be “were”
Please we have affected the correction, and we are very grateful for pointing this weakness to us.
- Line 729: it should be “The results of this study” or just “This study”
Please it is well noted and have been corrected. Many t hanks for pointing this weakness to us.
- The conclusion is too long and should be only one paragraph as far as I am concerned.
Please we have revised the conclusion
- Authors should carefully check throughout the manuscript for grammar and writing.
We are very grateful for pointing this weakness to us. We have proofread to correct the issues with grammar.
Comments on the Quality of English Language
Authors should carefully check throughout the manuscript for grammar and writing.
Again, thank you very for taking your time to review this paper.

Round 2
Reviewer 1 Report
I am pleased to note that the authors have made substantial improvements to their work based on my initial feedback, and I believe this study holds significant merit for publication in Agronomy.
The authors have addressed many of my concerns and made notable amendments to the manuscript. However, only one key issue still require attention, and I offer the following comments for your consideration:
ü I appreciate your willingness to consider my recommendation of complementing your trait-by-trait analyses with correlation analysis and Principal Component Analysis (PCA).
Author Response
Dear Reviewer:
Greetings. We are grateful for your review comments and suggestions. They have helped to improve upon the quality of the paper.
The reviewer recommended “ trait-by-trait analyses with correlation analysis and Principal Component Analysis (PCA)”.
We have performed the principal component analysis and the correlation on the data.
Please kindly find the revised manuscript attached for your consideration.
Thank you
Regards

Reviewer 3 Report
Dear Authors,
Please find my comments in the uploaded file.

Author Response
Dear Reviewer:
Greetings. We are grateful for your review comments because it has helped to improve upon the quality of the paper.
Please kindly find our response below for your consideration.
Thank you
Regards
